# DPVIm: Differentially Private Variational Inference Improved

**Joonas Jälkö**[*]                                                   *joonas.jalko@helsinki.fi*
*Department of Computer Science, University of Helsinki*

**Lukas Prediger**                                                   *lukas.m.prediger@aalto.fi*
*Department of Computer Science, Aalto University*

**Antti Honkela**                                                    *antti.honkela@helsinki.fi*
*Department of Computer Science, University of Helsinki*

**Samuel Kaski**                                                     *samuel.kaski@aalto.fi*
*Department of Computer Science, Aalto University*
*Department of Computer Science, University of Manchester*

**Reviewed on OpenReview:** *https://openreview.net/forum?id=GlhM6XX1wv*

## Abstract

Differentially private (DP) release of multidimensional statistics typically considers an aggregate sensitivity, e.g. the vector norm of a high-dimensional vector. However, different dimensions of that vector might have widely different magnitudes and therefore DP perturbation disproportionately affects the signal across dimensions. We observe this problem in the gradient release of the DP-SGD algorithm when using it for variational inference (VI), where it manifests in poor convergence as well as high variance in outputs for certain variational parameters, and make the following contributions: (i) We mathematically isolate the cause for the difference in magnitudes between gradient parts corresponding to different variational parameters. Using this as prior knowledge we establish a link between the gradients of the variational parameters, and propose an efficient while simple fix for the problem to obtain a less noisy gradient estimator, which we call *aligned* gradients. This approach allows us to obtain the updates for the covariance parameter of a Gaussian posterior approximation without a privacy cost. We compare this to alternative approaches for scaling the gradients using analytically derived preconditioning, e.g. natural gradients. (ii) We suggest using iterate averaging over the parameter iterates recovered during the training, to reduce the DP-induced noise in parameter estimates at no additional cost in privacy. Finally, (iii) to accurately capture the additional uncertainty DP introduces to the model parameters, we infer the DP-induced noise from the parameter iterates and include that in the learned posteriors to make them *noise aware*. We demonstrate the efficacy of our proposed improvements through various experiments on real data.

## 1 Introduction

Differential privacy (DP) (Dwork et al., 2006) protects privacy of data subjects by limiting how much about the input data can be learned from the output of an algorithm. Additive noise mechanisms achieve DP by adding noise calibrated to the maximum change in function output due to a single individual, known as sensitivity. When releasing high-dimensional data through such mechanisms, different variables may have widely different sensitivities. However, this issue of varying sensitivities is often neglected or overlooked.

---

[*]Work done while at Aalto University

Instead, the sensitivity of the release is computed as an aggregate over all the dimensions, which we call *total sensitivity*, in contrast to *variable-specific sensitivity*. As the DP noise is subsequently scaled with this total sensitivity, it affects dimensions with lower sensitivities more. A prominent example where this occurs is the gradient release in DP stochastic gradient descent (DP-SGD) (Song et al., 2013; Bassily et al., 2014; Abadi et al., 2016), where it can affect the convergence rate of the corresponding parameters. Furthermore, the final parameters released from DP-SGD are noisy estimators of the optimal parameters and the resulting error is usually treated as an unavoidable trade-off of providing privacy (Abadi et al., 2016). As a result, final parameter estimates with small variable-specific sensitivity may exhibit larger errors due to DP randomness compared to other dimensions. The combination of these two issues means that parameters with relatively small sensitivity are at a double disadvantage.

We discover these issues in the perturbed gradients used in the DP-SGD based DP variational inference (DPVI) algorithm (Jälkö et al., 2017), which is a widely applicable state-of-the-art method for privacy-preserving (approximate) Bayesian inference. We find that gradient magnitudes for different parameters in DPVI often differ significantly, resulting in severe errors in capturing the posterior. This results e.g. in poor predictive uncertainty estimation, making the predictions of the learned model less accurate.

We mathematically isolate the cause for these problems in DPVI and propose and evaluate two ways of alleviating the problem of gradient scales in DPVI: one scales gradients with a preconditioning matrix before applying the DP mechanism, the other is based on insights into the mathematical structure of the gradients, which reveals that their components are mathematically linked and can be derived from each other in a post-processing step.

Additionally, we theoretically and experimentally evaluate the method of iterate averaging as a way to further improve the parameter estimate as well as approximate the additional variance induced by DP perturbations to DPVI to make the posterior approximation noise aware at no additional cost in privacy.

## 1.1 Related work

In the context of DP-SGD, the following previous works acknowledge the different sensitivities of different parts of the full gradient: McMahan et al. (2018) suggested clipping the gradients of a neural network separately for each layer to avoid the clipping-induced bias (Chen et al., 2020). Other lines of work (Andrew et al., 2021; Wang et al., 2022) suggest adaptive clipping, where the total sensitivity is re-evaluated throughout the optimisation process to avoid adding excessive amounts of noise to the gradients. However, since in all these the perturbation is still scaled with the total sensitivity aggregated over the dimensions, this approach does not improve the disparate effect that the Gaussian noise will have on the dimensions with smaller gradients, so we see these approaches as orthogonal to our work. Besides the aforementioned works that study the tuning of clipping threshold, there are some recent works that study the use of clipping in terms of obtaining optimal rates in DP convex optimization Kamath et al. (2022); Lowy & Razaviyayn (2023).

For noise aware DP Bayesian inference, the most related work is by Bernstein and Sheldon (Bernstein & Sheldon, 2018; 2019) and Kulkarni et al. (2021). These works include the DP perturbation mechanism into a probabilistic model using perturbed sufficient statistics as the inputs. This allows capturing the DP-induced additional uncertainty in the posterior distribution of model parameters.

## 2 Preliminaries

### 2.1 Differential privacy

**Definition 2.1** (Differential Privacy (Dwork et al., 2006)). For $\epsilon \geq 0$ and $\delta \in [0, 1]$, a randomised mechanism $\mathcal{M} : \mathcal{D} \to \mathcal{R}$ satisfies $(\epsilon, \delta)$-differential privacy if for any two data sets different in only one element, $D, D' \in \mathcal{D}$, and for all outputs $S \subseteq \text{im}(\mathcal{M})$, the following constraint holds:

$$\Pr(\mathcal{M}(D) \in S) \leq e^{\epsilon} \Pr(\mathcal{M}(D') \in S) + \delta. \tag{1}$$

**Property 2.1** (Post-processing immunity (cf. (Dwork et al., 2014))). Let $\mathcal{M} : \mathcal{D} \to \mathcal{R}$ be a $(\epsilon, \delta)$-DP mechanism and $f : \mathcal{R} \to \mathcal{Z}$ any function that does not access the sensitive data. Then $f \circ \mathcal{M}$ is $(\epsilon, \delta)$-DP.

## 2.2 Variational inference

Variational inference is a commonly applied technique in probabilistic inference, where the aim is to learn an approximation for a (typically intractable) posterior distribution of the parameters of a probabilistic model (Jordan et al., 1999). The goal is to minimise the Kullback-Leibler (KL) divergence of this approximation to the true posterior. However, computing the KL divergence directly is intractable as well, so instead we maximise a quantity called *evidence lower bound* (ELBO) over the parameters of the variational approximation. For a probabilistic model $p(D, \boldsymbol{\theta})$, where $D$ denotes the observed variables and $\boldsymbol{\theta}$ the model parameters, and for a variational approximation $q(\boldsymbol{\theta})$ of the posterior, the ELBO $\mathcal{L}(q)$ is derived as follows:

$$\mathrm{KL}(q(\boldsymbol{\theta}) \,||\, p(\boldsymbol{\theta} \mid D)) = \mathbb{E}_{q(\boldsymbol{\theta})} \left[ \log \frac{q(\boldsymbol{\theta}) p(D)}{p(D, \boldsymbol{\theta})} \right] \tag{2}$$

$$= \log p(D) - \underbrace{\mathbb{E}_{q(\boldsymbol{\theta})} \left[ \log \frac{p(D, \boldsymbol{\theta})}{q(\boldsymbol{\theta})} \right]}_{:=\mathcal{L}(q)} \tag{3}$$

Now, as the KL divergence is positive we have

$$\log p(D) \geq \mathcal{L}(q), \tag{4}$$

hence the quantity $\mathcal{L}(q)$ is called the evidence lower bound. We can easily see that if the KL divergence between the $q$ and the posterior is 0, the ELBO matches the evidence. Since the evidence is independent of the variational parameters, minimizing the KL divergence w.r.t the variational parameters is equivalent to maximizing the ELBO.

In the remainder of this paper we use the following equivalent formulation of the ELBO:

$$\mathcal{L}(q) = \mathbb{E}_{q(\boldsymbol{\theta})} \left[ \log p(D, \boldsymbol{\theta}) \right] + H(q), \tag{5}$$

where $H(q)$ denotes the (differential) entropy of $q$.

In the following we first restrict ourselves to the commonly used *mean-field* variational inference, i.e., using a Gaussian with diagonal covariance as the posterior approximation. We will later generalise this to a full-rank coveriance approximation. For $d$-dimensional data the diagonal approximation is parametrised by the means $\boldsymbol{m}_q \in \mathbb{R}^d$ and the dimension-wise standard deviations $\boldsymbol{\sigma}_q \in \mathbb{R}^d$. We further reparametrise the model with $\boldsymbol{s}_q = T^{-1}(\boldsymbol{\sigma}_q)$, where $T : \mathbb{R} \to \mathbb{R}_+$ is monotonic, in order to facilitate optimisation in an unconstrained domain. Both $T$ and $T^{-1}$ are applied element-wise for each of the parameters. Common choices for $T$ are the exponential function $T(s) = \exp(s)$ and the softplus function $T(s) = \log(\exp(s) + 1)$ (used e.g. in the Pyro probabilistic programming package (Bingham et al., 2019)). We use $\boldsymbol{\xi} = (\boldsymbol{m}_q, \boldsymbol{s}_q)$ to refer to the complete set of variational parameters.

A draw from this posterior distribution can then be written as (Kingma & Welling, 2014):

$$\boldsymbol{\theta} := \boldsymbol{\theta}(\boldsymbol{\eta}; \boldsymbol{m}_q, \boldsymbol{s}_q) = \boldsymbol{m}_q + T(\boldsymbol{s}_q)\boldsymbol{\eta}, \tag{6}$$

where $\boldsymbol{\eta} \sim N(0, I_d)$, $\boldsymbol{\theta}, \boldsymbol{\eta} \in \mathbb{R}^d$ and $I_d$ is a $d$-dimensional identity matrix. Kucukelbir et al. (2017) use this reparametrisation trick together with single-sample MC integration to give the ELBO a differentiable form with gradients:

$$\boldsymbol{g}_m := \nabla_{\boldsymbol{m}_q} \mathcal{L}(q) = \nabla_{\boldsymbol{m}_q} \log p(D, \boldsymbol{\theta}(\boldsymbol{\eta}; \boldsymbol{m}_q, \boldsymbol{s}_q)) \tag{7}$$

$$\boldsymbol{g}_s := \nabla_{\boldsymbol{s}_q} \mathcal{L}(q) = \nabla_{\boldsymbol{s}_q} \log p(D, \boldsymbol{\theta}(\boldsymbol{\eta}; \boldsymbol{m}_q, \boldsymbol{s}_q)) + \nabla_{\boldsymbol{s}_q} H(q), \tag{8}$$

where $\boldsymbol{\eta} \sim N(0, I)$. Throughout this work we assume that the likelihood factorises as: $p(D \mid \boldsymbol{\theta}) = \prod_{\boldsymbol{x} \in D} p(\boldsymbol{x} \mid \boldsymbol{\theta})$. Using $N$ to denote the size of $D$, we can now further decompose the gradients in (7) and (8) as

$$\boldsymbol{g}_m = \sum_{\boldsymbol{x} \in D} \left( \nabla_{\boldsymbol{m}_q} \log p(\boldsymbol{x} \mid \boldsymbol{\theta}(\boldsymbol{\eta}; \boldsymbol{m}_q, \boldsymbol{s}_q)) + \frac{1}{N} \nabla_{\boldsymbol{m}_q} \log p(\boldsymbol{\theta}(\boldsymbol{\eta}; \boldsymbol{m}_q, \boldsymbol{s}_q)) \right) \tag{9}$$

$$\boldsymbol{g}_s = \sum_{\boldsymbol{x} \in D} \left( \nabla_{\boldsymbol{s}_q} \log p(\boldsymbol{x} \mid \boldsymbol{\theta}(\boldsymbol{\eta}; \boldsymbol{m}_q, \boldsymbol{s}_q)) + \frac{1}{N} \left( \nabla_{\boldsymbol{s}_q} \log p(\boldsymbol{\theta}(\boldsymbol{\eta}; \boldsymbol{m}_q, \boldsymbol{s}_q)) + \nabla_{\boldsymbol{s}_q} H(q) \right) \right). \tag{10}$$

We denote the per-example gradient components (i.e., those for each individual $\boldsymbol{x}$) that appear in the above sums with $\boldsymbol{g}_{m,\boldsymbol{x}}$ and $\boldsymbol{g}_{s,\boldsymbol{x}}$ respectively.

A common approach to performing variational inference in practice is to initialise $\boldsymbol{s}_q$ to small values, which allows the algorithm to move $\boldsymbol{m}_q$ quickly close to their optimal values due to large error in the KL term of the ELBO induced by the narrow approximation.

**Assumptions for the probabilistic model and variational posterior**   For the rest of the paper, we make the following assumptions:

- **Exchangeability**: We assume that there is no ordering in observing the elements of data set $D$: $p(D \mid \boldsymbol{\theta}) = \prod_{x \in D} p(\mathbf{x} \mid \boldsymbol{\theta})$

- **Gaussian posterior approximation**: $q(\boldsymbol{\theta}) = \mathcal{N}_d(\boldsymbol{m}_q, \boldsymbol{\Sigma}_q)$, where $\mathcal{N}_d$ denotes the pdf of a $d$-dimensional Gaussian. When working with isotropic Gaussian posterior approximation (i.e. diagonal covariance), we denote the dimension-wise standard deviations with $\boldsymbol{\sigma}_q$.

- **Optimising the $\boldsymbol{\sigma}_q$**: For the isotropic Gaussian, we use a mapping function $T : \mathbb{R} \to \mathbb{R}^+$ to optimize the variational standard deviations. We apply this function element-wise to the parameter vector $\boldsymbol{s}_q$. Commonly used examples for this are the softplus function $T(s) = \log(1 + \exp(s))$ as well as the exponential function $T(s) = \exp(s)$.

## 3   Differentially private variational inference

The first algorithm for differentially private variational inference for non-conjugate models (Jälkö et al., 2017) optimises the ELBO using gradients (7) and (8) with differentially private stochastic gradient descent (Abadi et al., 2016) to provide privacy. This involves concatenating each of the per-example gradients to obtain $\boldsymbol{g}_{\boldsymbol{x}} = (\boldsymbol{g}_{m,\boldsymbol{x}}^T, \boldsymbol{g}_{s,\boldsymbol{x}}^T)^T$, clipping $\boldsymbol{g}_{\boldsymbol{x}}$ so that it has $\ell_2$ norm no larger than a threshold $C$ to limit the sensitivity, and finally adding Gaussian noise to the sum of these clipped per-example gradients to obtain $\tilde{\boldsymbol{g}}$, which is used for the parameter update. We refer to this algorithm in the following as *vanilla DPVI*.

This formulation induces a problem which, while seemingly minor at first glance, severely affects accuracy of solutions. We next isolate this problem and then propose a solution through detailed analysis on the gradients of the variational parameters.

### 3.1   Disparate perturbation of variational parameter gradients

While the clipping of the gradients allows us to bound the global sensitivity of the gradient vector, it completely ignores any differences in gradient magnitudes across the dimensions. As DPVI (and more generally DP-SGD) proceeds to add Gaussian noise with standard deviation proportional to the clipping threshold to *all of the dimensions*, the signal-to-noise-ratio can vary greatly across the parameter dimensions. Parameter dimensions that experience low signal-to-noise ratio will converge much slower than others (cf. Domke, 2019 and references therein).[1]  Next, we will show that such a magnitude difference arises between the gradients of variational parameters $\boldsymbol{m}_q$ and $\boldsymbol{s}_q$.

Note that the gradient of Equation (6) w.r.t. $\boldsymbol{m}_q$ is $\nabla_{\boldsymbol{m}_q} \boldsymbol{\theta}(\boldsymbol{\eta}; \boldsymbol{m}_q, \boldsymbol{s}_q) = 1$, which leads to the following proposition (a more detailed derivation can be found in Appendix B):

**Proposition 3.1.** Assume $q$ to be diagonal Gaussian, then the gradient $\boldsymbol{g}_s$ in Equation (8) becomes

$$\boldsymbol{g}_s = \boldsymbol{\eta} T'(\boldsymbol{s}_q) \boldsymbol{g}_m + \nabla_{\boldsymbol{s}_q} H(q), \tag{11}$$

where $T'$ denotes the derivative of $T$.

As the entropy term is independent of the data, our update equation for $\boldsymbol{s}_q$ depends on the data only through $\boldsymbol{g}_m$. In order to show that this term gets affected by the noise more than $\boldsymbol{g}_m$ itself, it suffices to inspect

---

[1]We also provide a high-level argument why this is the case in Appendix A.

the magnitudes of $\boldsymbol{\eta}$ and $T'(\boldsymbol{s}_q)$. We have $\Pr\big[|\boldsymbol{\eta}_j| < 2\big] > 97.5\%$ and $T'(\boldsymbol{s}_q) \leq \boldsymbol{\sigma}_q$ for common choices of $T$ discussed above (a proof for softplus and exp can be found in Appendix C). For each dimension of the data dependent part of $\boldsymbol{g}_s$ it then follows that

$$|\boldsymbol{\eta}_j T'(\boldsymbol{s}_{q,j})\boldsymbol{g}_{m,j}| \leq \boldsymbol{\sigma}_{q,j}|\boldsymbol{\eta}_j \boldsymbol{g}_{m,j}| < 2\boldsymbol{\sigma}_{q,j}|\boldsymbol{g}_{m,j}| \text{ with probability} > 97.5\%. \tag{12}$$

Therefore, it is easy to see that as $\boldsymbol{\sigma}_q$ becomes small, the data-dependent part of the gradient $\boldsymbol{g}_s$ becomes small compared to $\boldsymbol{g}_m$ in the overwhelming majority of cases. The entropy part of $\boldsymbol{g}_s$ is typically small as well, especially when we are far from the posterior mode, i.e., early in training. As a result, $\boldsymbol{g}_s$ as a whole becomes small in comparison to $\boldsymbol{g}_m$. Note that this is especially problematic combined with the practice of initialising $\boldsymbol{s}_q$ to small values to speed up the convergence of $\boldsymbol{m}_q$, discussed in Sec. 2.2.

### 3.1.1 Addressing the differing gradient scales

Above we have identified a magnitude difference between the gradient components, which leads to variational standard deviation parameters being disproportionately affected by DP noise. Next we use this structural knowledge to propose a method for scaling the gradients to more closely matching magnitudes, after discussing two alternatives based on standard techniques.

**Natural Gradients** As a first solution, we consider *natural gradients* (Amari, 1998). This is a common approach for improving convergence for VI (cf. e.g. Honkela et al., 2010; Khan & Nielsen, 2018; Salimbeni et al., 2018), which relies on scaling the gradients using the information geometry of the optimisation problem.

The natural gradients $\boldsymbol{g}^{nat}$ are computed using the inverse of the Fisher information matrix $\mathcal{I}$ as

$$\mathcal{I} = \mathbb{E}_{\boldsymbol{\theta}|\boldsymbol{s}_q,\boldsymbol{m}_q}\left[(\nabla_{\boldsymbol{\theta}}\log q(\boldsymbol{\theta}))(\nabla_{\boldsymbol{\theta}}\log q(\boldsymbol{\theta}))^T\right] \tag{13}$$

$$\boldsymbol{g}^{nat} = \mathcal{I}^{-1}\boldsymbol{g}. \tag{14}$$

For our setting this leads to

$$\boldsymbol{g}_m^{nat} = T(\boldsymbol{s}_q)^2 \nabla_{\boldsymbol{m}_q}\mathcal{L}(q) \tag{15}$$

$$\boldsymbol{g}_s^{nat} = \frac{1}{2T'(\boldsymbol{s}_q)}\left(\boldsymbol{\eta}\boldsymbol{g}_m^{nat} + \frac{T(\boldsymbol{s}_q)^2}{T'(\boldsymbol{s}_q)}\nabla_{\boldsymbol{s}_q}H(q)\right). \tag{16}$$

We observe that in the natural gradients the scaling by $T'(\boldsymbol{s}_q)$ in the gradients of $\boldsymbol{s}_q$ is now reversed, meaning that for small $T'(\boldsymbol{s}_q)$ the gradients of $\boldsymbol{s}_q$ will tend to dominate over those of $\boldsymbol{m}_q$. Therefore we expect natural gradients to result in a different instance of the problem of disproportionate DP noise instead of resolving it.

**Preconditioning of Gradients** The simplest way to fix the disproportionate DP noise is preconditioning of the gradients to undo the downscaling of the data-dependent part in Eq. (11), by multiplying with $(T'(\boldsymbol{s}_q))^{-1}$, to obtain

$$\boldsymbol{g}_s^{precon} = \frac{1}{T'(\boldsymbol{s}_q)}\boldsymbol{g}_s = \boldsymbol{\eta}\boldsymbol{g}_m + \frac{\nabla_{\boldsymbol{s}_q}H(q)}{T'(\boldsymbol{s}_q)}. \tag{17}$$

We can see that the data dependent part of $\boldsymbol{g}_s^{precon}$ (the first term) is of the same magnitude as $\boldsymbol{g}_m$, and thus the noise affects the gradient components equally.[2] Note that while this approach addresses the issue of different magnitude in the gradients, it does so at the cost of increasing the overall $\ell_2$-norm of the full gradient (by increasing that of $\boldsymbol{g}_s$ while keeping $\boldsymbol{g}_m$ fixed). This in turn requires a higher clipping threshold in order to avoid additional bias due to clipping, which increases DP noise variance.

---

[2]Note that this scaling also affects the data-independent entropy term in the gradient for $\boldsymbol{s}_q$. While the scaling term $(T'(\boldsymbol{s}_q))^{-1}$ does increase the entropy part for small $\boldsymbol{s}_q$, the data-dependent term is still typically much larger and will dominate the gradient.

**Aligned Gradients** We will now discuss a new alternative method for resolving the disproportionate DP noise problem that addresses the gradient magnitude problem while avoiding the issues of the preconditioning approach. Equation (11) shows that we can write $\boldsymbol{g}_s$ in terms of $\boldsymbol{g}_m$ and an additional entropy term. Since neither the scaling factor $\boldsymbol{\eta} T'(\boldsymbol{s}_q)$ nor the entropy gradient $\nabla_{\boldsymbol{s}_q} H(q)$ depend on the data $D$, it suffices to release the gradients $\boldsymbol{g}_m$ under DP as $\tilde{\boldsymbol{g}}_m$, from which we obtain the $\tilde{\boldsymbol{g}}_s$ via Eq. (11). As this is simply post-processing, it does not incur additional DP cost. Because $\tilde{\boldsymbol{g}}_s$ is now computed directly as a transformation of $\tilde{\boldsymbol{g}}_m$, the noise term in both gradients is aligned in proportion to the gradient signals. We refer to this approach as *aligned DPVI* for the rest of the paper. The procedure for computing the aligned DPVI gradients is summarized in Algorithm 1.

---

**Algorithm 1** The aligned gradient procedure  (single step)

---

1: $\boldsymbol{\theta} \leftarrow \boldsymbol{m}_q + \boldsymbol{\eta} T(\boldsymbol{s}_q)$ where $\boldsymbol{\eta} \sim \mathcal{N}(0, I)$ $\qquad\qquad$ ▷ Draw sample from the variational posterior
2: $\boldsymbol{g}_{m,\boldsymbol{x}} \leftarrow \nabla_{\boldsymbol{m}_q} \mathcal{L}(q)$ for $\boldsymbol{x} \in D$ $\qquad\qquad\qquad$ ▷ Compute the per-example gradients for $\boldsymbol{m}_q$
3: $\gamma_{\boldsymbol{x}} \leftarrow \min(1, C/\|\boldsymbol{g}_{m,\boldsymbol{x}}\|)$ for $\boldsymbol{x} \in D$ $\qquad\qquad\qquad\qquad$ ▷ Compute the clipping multiplier
4: $\tilde{\boldsymbol{g}}_m \leftarrow \sum_{\boldsymbol{x} \in D} \gamma_{\boldsymbol{x}} \boldsymbol{g}_{m,\boldsymbol{x}} + \sigma_{DP} C \boldsymbol{\psi}$, where $\boldsymbol{\psi} \sim \mathcal{N}(0, I)$ $\qquad\qquad$ ▷ Get DP release for $\boldsymbol{g}_m$
5: $\tilde{\boldsymbol{g}}_s^{aligned} \leftarrow \boldsymbol{\eta} T'(\boldsymbol{s}_q) \tilde{\boldsymbol{g}}_m + \nabla_{\boldsymbol{s}_q} H(q)$ $\qquad\qquad$ ▷ Get DP aligned $\boldsymbol{g}_s$ via post-processing

---

**Corrolary 3.1.** Let $\delta(\epsilon; \sigma_{DP}, T)$ be any privacy accounting oracle. Let $\epsilon > 0$, $\sigma_{DP} > 0$. Then aligned DPVI consisting of $T$ iterations of Alg. 1 is $(\epsilon, \delta(\epsilon; \sigma_{DP}, T))$-DP.

*Proof.* The $\tilde{\boldsymbol{g}}_m$ released in step 4 of Algorithm 1 satisfies the privacy guarantee provided by the Gaussian mechanism. As the $\boldsymbol{\eta}$, $T'(\boldsymbol{s}_q)$ and the entropy contribution $\nabla_{\boldsymbol{s}_q} H(q)$ are independent of the data, the $\tilde{\boldsymbol{g}}_s^{aligned}$ satisfies the same privacy guarantee through post-processing immunity (Property 2.1). Hence the privacy guarantees of the entire algorithm follow from the composed privacy cost of the released gradients for the variational means $\{\tilde{\boldsymbol{g}}_m^{(t)}\}_{t=1}^T$. $\qquad\qquad\qquad\qquad\qquad\qquad\qquad\qquad\qquad\qquad\qquad\qquad\quad$ □

The following theorem (proved in Appendix D) guarantees that the variance in the gradients of $\boldsymbol{s}_q$ is reduced in aligned DPVI:

**Theorem 3.1.** Assume that $C^{vanilla} \geq C^{aligned}$. If we obtain $\boldsymbol{\sigma}_q$ through transformation $T$ such that $T'(s) \leq 1$, then for any fixed batch,

$$\mathrm{Var}_{\boldsymbol{\eta},\boldsymbol{\psi}} \left[ \tilde{\boldsymbol{g}}_s^{aligned} \right] < \mathrm{Var}_{\boldsymbol{\eta},\boldsymbol{\psi}} \left[ \tilde{\boldsymbol{g}}_s^{vanilla} \right], \tag{18}$$

where $\boldsymbol{\eta}$ is the random variable of the MC approximation to the ELBO and $\boldsymbol{\psi}$ that of the DP perturbation.

Note that the softplus transformation used in our experiments satisfies the assumption $T'(s) \leq 1$.

**Aligned Natural Gradients** Finally we also consider a combination of natural gradients and aligning, to enable the benefits of natural gradients for convergence while simultaneously removing the need to consider the gradient of $\boldsymbol{s}_q$ for DP clipping and perturbation. The full procedure is given in Algorithm 2. We use $\mathcal{I}_m, \mathcal{I}_s$ to refer to the blocks of $\mathcal{I}$ corresponding to the gradient components.

---

**Algorithm 2** The aligned natural gradient procedure (single step)

---

1: $\boldsymbol{\theta} \leftarrow \boldsymbol{m}_q + \boldsymbol{\eta} T(\boldsymbol{s}_q)$ where $\boldsymbol{\eta} \sim \mathcal{N}(0, I)$ $\qquad\qquad\qquad$ ▷ Draw sample from the variational posterior
2: $\boldsymbol{g}_{m,\boldsymbol{x}}^{nat} \leftarrow \mathcal{I}_m^{-1} \nabla_{\boldsymbol{m}_q} \mathcal{L}(q)$ for $\boldsymbol{x} \in D$ $\qquad\qquad$ ▷ Compute per-example natural gradients for $\boldsymbol{m}_q$
3: $\gamma_{\boldsymbol{x}} \leftarrow \min(1, C/\|\boldsymbol{g}_{m,\boldsymbol{x}}^{nat}\|)$ for $\boldsymbol{x} \in D$ $\qquad\qquad\qquad\qquad$ ▷ Compute the clipping multiplier
4: $\tilde{\boldsymbol{g}}_m^{nat} \leftarrow \sum_{\boldsymbol{x} \in D} \gamma_{\boldsymbol{x}} \boldsymbol{g}_{m,\boldsymbol{x}} + \sigma_{DP} C \boldsymbol{\psi}$, where $\boldsymbol{\psi} \sim \mathcal{N}(0, I)$ $\qquad\qquad$ ▷ Get DP release for $\boldsymbol{g}_m^{nat}$
5: $\tilde{\boldsymbol{g}}_s^{nat,aligned} = \mathcal{I}_s^{-1} (\boldsymbol{\eta} T'(\boldsymbol{s}_q) \mathcal{I}_m \tilde{\boldsymbol{g}}_m^{nat} + \nabla_{\boldsymbol{s}_q} H(q))$. $\qquad$ ▷ Get DP aligned $\boldsymbol{g}_s^{nat}$ via post-processing

---

### 3.1.2 Extending to full-rank covariance matrices

So far we have only considered a diagonal Gaussian as the variational posterior. Due to the low dimensionality of the variational parameters, this approach is computationally effective and often applied in practice, but it

has limitations: It cannot capture correlations among different model parameters and, more importantly, it will underestimate the marginal variances of the parameters when the true covariance structure is non-diagonal. For those reasons, a full-rank covariance approximation would be favored to correctly capture the uncertainty of the parameters.

However, learning the full-rank covariance approximation results in a quadratic (in the number of dimensions $d$) expansion of the number of learnable parameters. This not only increases computational costs but also implies less accurate learning of the parameters under DP, as the available privacy budget has to be spread over more parameters. Fortunately, the aligning procedure can be extended to full-rank Gaussian approximations as well, which allows us to alleviate the issue of increased sensitivity. The proof is very similar to the diagonal case. Instead of the parameters $\boldsymbol{s}_q$ corresponding to marginal standard deviations, we now consider a parameter vector $\boldsymbol{a}_q \in \mathbb{R}^{\frac{d(d+1)}{2}}$ and a transformation function $T : \mathbb{R}^{\frac{d(d+1)}{2}} \to \mathbb{R}^{d \times d}$ such that $T(\boldsymbol{a}_q)$ corresponds to the Cholesky factor of the posterior covariance. That is, $T$ must guarantee that $T(\boldsymbol{a}_q)$ is a lower triangular with positive entries along its diagonal, which will require similar transformations as in the purely diagonal covariance case discussed previously. Now, the reparametrisation step in (6) becomes

$$\theta := \boldsymbol{m}_q + T(\boldsymbol{a}_q)\boldsymbol{\eta}, \tag{19}$$

and the gradient w.r.t $\boldsymbol{a}_q$ can be written as

$$\boldsymbol{g}_a = J_a(T(\boldsymbol{a}_q)\boldsymbol{\eta})\boldsymbol{g}_m + \nabla_{\boldsymbol{a}_q} H(q), \tag{20}$$

where $J_a$ denotes the Jacobian of $T$ w.r.t $\boldsymbol{a}_q$. Therefore, the gradient $\boldsymbol{g}_a$ can again be written as a data-independent transformation of $\boldsymbol{g}_m$. Thus under the post-processing immunity of DP, we can get the DP gradients for $\boldsymbol{a}_q$ from DP versions of $\boldsymbol{g}_m$ without suffering the quadratic increase of the size of the input to underlying the Gaussian mechanism present in vanilla DPVI.

### 3.2 Leveraging DP parameter iterates to reduce error and for uncertainty estimation

As other applications of DP-SGD, vanilla DPVI does not take the uncertainty that the DP mechanism introduces into account. Instead, after a finite number of iterations, the values found in the last iteration are usually treated as the true variational parameters. We argue that treating them this way can lead to severe errors in accuracy because these values are merely a noisy estimate of the optimal values. Annealing the learning rate to reduce fluctuations around the optimum does not help: Without knowledge of the convergence point the distance to an optimum can still be large due to the random walk prior to annealing. Instead in the following we suggest making use of some fraction of the DP parameter iterates output by the algorithm to 1) average out and 2) estimate the additional variance introduced by DP noise additions, making the learned model approximately noise-aware. While 1) is a known technique known as iterate averaging (Polyak & Juditsky, 1992), 2) has not been previously applied in this context to the best of our knowledge. Note that privacy guarantees of DP-SGD algorithms extend to all parameter iterates, so this comes at no additional privacy cost. We first briefly review some theory about the random walk behavior around the optimum and then discuss the details of our proposed approach.

Mandt et al. (2017) investigated the random walk behaviour around the optimum for regular (non-DP) SGD arising from subsampling. They assume that near the optimum $\boldsymbol{\xi}^*$ the loss function is well approximated by a quadratic approximation $L(\boldsymbol{\xi}) \approx \frac{1}{2}(\boldsymbol{\xi} - \boldsymbol{\xi}^*)^T \boldsymbol{A}(\boldsymbol{\xi} - \boldsymbol{\xi}^*)$, and show that the stochastic process around the optimum can be characterised as an Ornstein-Uhlenbeck (OU) process

$$d\boldsymbol{\xi}(t) = -\alpha \boldsymbol{A}(\boldsymbol{\xi}(t) - \boldsymbol{\xi}^*)dt + \frac{1}{\sqrt{S}}\alpha \boldsymbol{B}dW(t), \tag{21}$$

where $W(t)$ is a Wiener process, $\alpha$ is the step size of the SGD (assumed constant), $S$ is the size of the subsampled data and $\boldsymbol{B}$ the Cholesky decomposition of the covariance matrix $\boldsymbol{Z}$ of the noise due to the subsampling. Directly adapting this analysis, we suggest that under the same regularity assumptions, DP-SGD still is an OU process. The principle of the proof is straightforward: DP-SGD adds an additional Gaussian noise component, allowing us to add the (diagonal) covariance matrix of the DP noise to $\boldsymbol{Z}$ and obtain a $\hat{\boldsymbol{B}}$

such that

$$\boldsymbol{Z} + S\sigma_{DP}^2\boldsymbol{I} = \hat{\boldsymbol{B}}\hat{\boldsymbol{B}}^T. \tag{22}$$

The more detailed proof can be found in Appendix E. This insight allows us to make the following two suggestions to improve the parameter estimates of DPVI.

**Iterate averaging to reduce noise in parameter estimate**  In order to reduce noise in our learned variational parameters, we apply iterate averaging and average the parameter traces, i.e., the sequence of parameter iterates during optimisation, over the last $T_{\text{burn-out}}$ iterates for which we assume the trace has converged. As the OU process in Equation (21) is symmetric around the optimum, the mean of the trace is an unbiased estimator of $\boldsymbol{\xi}^*$. Compared to using the final iterate of the chain (also an unbiased estimator of $\boldsymbol{\xi}^*$), the averaged trace reduces the variance of the estimator by up to a factor of $T_{\text{burn-out}}^{-1}$. Iterate averaging has been previously been used to reduce the noise of DP-SGD for example by Bassily et al. (2014) by scaling the learning rate w.r.t total number of iterations and by Bassily et al. (2019) and Lowy & Razaviyayn (2021) by taking average over all the iterates obtained during training. However, the key difference in our method is that we are actively estimating the point of convergence for the iterates, and only averaging after that. The convergence is required for estimating the DP induced noise from the OU process.

**Estimating the increased variance due to DP**  Finally, since our posterior approximation is Gaussian and the stationary distribution of the OU is Gaussian as well, we can add the variance of the averaged traces to the variances of our posterior to absorb the remaining uncertainty due to the inference process, and recover a noise-aware posterior approximation.

Now the remaining problem is to determine $T_{\text{burn-out}}$, the length of the trace where the parameters have converged. For this we suggest a simple convergence check based on linear regression: For each of the traces, we fit linear regression models over different candidate $T_{\text{burn-out}}$. The regressor $\boldsymbol{X}_{linreg}$ is set to interval $[0, 1]$ split to $T_{\text{burn-out}}$ points in an ascending order. The responses $y$ are set to the corresponding parameter values in the trace, e.g. $\mathbf{y} = \{\boldsymbol{m}_q^{(t)}\}_{t=T-T_{burn-out}}^T$. If the linear regression model has a sufficiently small slope coefficient, we consider the trace as converged and pick the longest $T_{\text{burn-out}}$ for which this is the case.

## 4 Experiments

We experimentally test our methods for two different tasks using mean-field approximation with real data: learning a probabilistic generative model for private data sharing and learning a logistic regression model. We also experimentally explore aligned DPVI with full-rank Gaussian approximation using simulated data.

### 4.1 Implementation details

We implemented the different variants for DPVI introduced in Sec. 3.1.1 using the d3p package (Prediger et al., 2022) for the NumPyro probabilistic programming framework (Phan et al., 2019; Bingham et al., 2019). To compute the privacy cost in our experiments, we use the Fourier accountant method (Koskela et al., 2021). The hyperparameters used in our experiments are discussed in the Appendix F. We use the softplus function as our transformation $T$ in all experiments. The code for reproducing the experiments can be found at `https://github.com/DPBayes/dpvim-experiments`.

In order to asses learning over multiple parameters which converge to different values, and over repeated runs with different initial values, we define a *mean proportional absolute error (MPAE)*: Let $\boldsymbol{\xi}^{(t)} \in \mathbb{R}^D$ be the parameter vector at iteration $t$ and $\boldsymbol{\xi}^*$ be the parameter vector at the optimum.[3] We measure the MPAE at iteration $t$ as

$$\text{MPAE}(\boldsymbol{\xi}^{(t)}) = \frac{1}{D}\sum_{d=1}^D \frac{|\boldsymbol{\xi}_d^{(t)} - \boldsymbol{\xi}_d^*|}{|\boldsymbol{\xi}_d^{(0)} - \boldsymbol{\xi}_d^*|}. \tag{23}$$

An MPAE value of 0 indicates perfect recovery of the optimum, a value of 1 suggests that the parameters on average did not move away from their initialisation.

---

[3]Since the optimal value $\boldsymbol{\xi}^*$ is typically unknown, we instead use the results of classical non-DP variational inference in its place in practice.

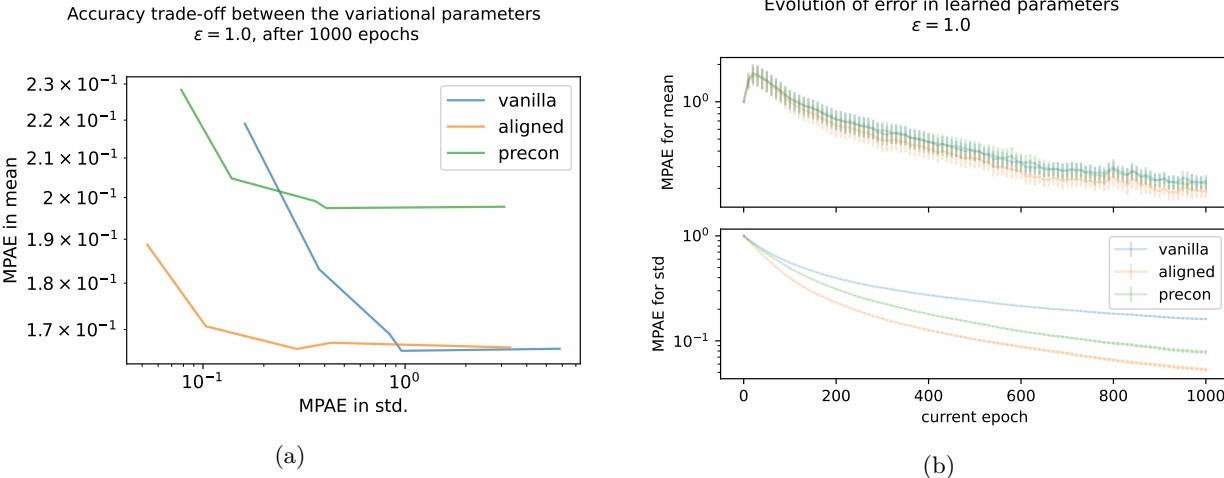

(a)  (b)

Figure 1: **UKB experiment:** (a) Aligned DPVI makes the smallest error in learning the variational parameters across all different initial values for $\boldsymbol{\sigma}_q$, implying it is the most robust. (b) Aligned DPVI converges faster than the other variants while also having less deviation across the repeats (all initialised at $\boldsymbol{\sigma}_q = 1$). Both subfigures show averaged MPAE for vanilla, aligned and preconditioned DPVI with error bars in (b) indicating standard error over repeats.

## 4.2 Using DPVI to learn a generative model

Recently, Jälkö et al. (2021) suggested using DPVI to learn a probabilistic generative model for differentially private data sharing. Note that in this application it is especially crucial to learn the posterior variances well to faithfully reproduce the uncertainty in the original data in the synthetic data set.

A recent study by Niedzwiedz et al. (2020) on personal health data from the United Kingdom Biobank (UKB) (Sudlow et al., 2015) studied how socio-economic factors affect an individual's risk of catching the SARS-CoV-2 virus. We aim to produce synthetic data, using DPVI to learn the generative model, from which we can draw similar discoveries.

Following Niedzwiedz et al. (2020), we consider a subset of UKB data which comprises of $58\,261$ individuals with $d = 7$ discrete (categorical) features. We split the features into a set of explanatory variables and a response variable indicating whether the individual was infected by SARS-CoV-2.

We place a mixture model for the explanatory variables $\boldsymbol{X}$, and a Poisson regression model mapping the explanatory variables to the responses $\boldsymbol{y}$, using $\boldsymbol{\theta_X}, \boldsymbol{\theta_y}$ and $\boldsymbol{\pi}$ to designate the model parameters:

$$p(\boldsymbol{X} \mid \boldsymbol{\theta_X}, \boldsymbol{\pi}) = \sum_{k=1}^{K} \pi_k \prod_{j=1}^{d} \mathrm{Categorical}(\boldsymbol{X}_j \mid \boldsymbol{\theta}_{\boldsymbol{X}}^{(k)}) \tag{24}$$

$$p(\boldsymbol{y} \mid \boldsymbol{X}, \boldsymbol{\theta_y}) = \mathrm{Poisson}(\boldsymbol{y} \mid \exp(\boldsymbol{X}\boldsymbol{\theta_y})). \tag{25}$$

In our experiments, we set the number of mixture components $K = 16$ which was chosen based on internal tests. Priors for the model parameters are specified in Appendix G.1.

**Aligned DPVI is more robust to initialisation** We first demonstrate that aligned DPVI improves robustness to initialisation over vanilla and preconditioned DPVI. To do so we fix a privacy budget of $\varepsilon = 1$ and the number of passes over the entire data set, i.e., *epochs*, to 1000 and vary the initial value of $\boldsymbol{s}_q$ such that $\boldsymbol{\sigma}_q$ is one of $0.01, 0.032, 0.1, 0.316$ or $1$. We perform 10 repetitions with different random seeds over which we keep the initialisation of $\boldsymbol{s}_q$ fixed but initialise $\boldsymbol{m}_q$ randomly. We compute the MPAE over the parameters of the Poisson regression part in the model, which corresponds directly to the downstream prediction task we are ultimately interested in.

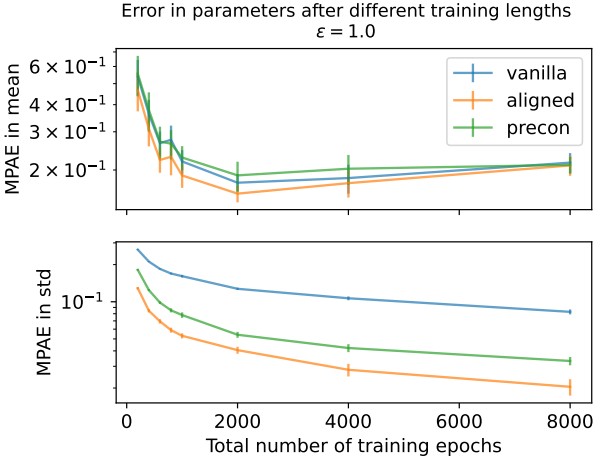

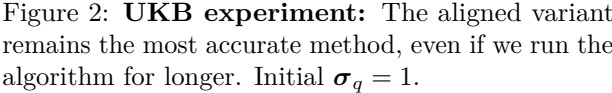

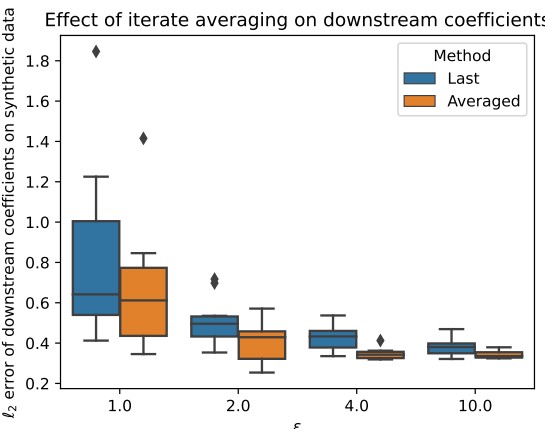

Figure 2: **UKB experiment:** The aligned variant remains the most accurate method, even if we run the algorithm for longer. Initial $\boldsymbol{\sigma}_q = 1$.

Figure 3: **UKB experiment:** RMSE of parameters found in downstream analysis when doing iterate averaging with different $T_{\text{burn-out}}$. Iterative averaging can reduce the error and significantly reduce variance of error compared to using only the last iterate. Initial $\boldsymbol{\sigma}_q = 1$, 4000 epochs of aligned DPVI.

Figure 1a shows the trade-off different variants of DPVI make between the MPAE in variational means ($\boldsymbol{m}_q$) and stds ($\boldsymbol{s}_q$) averaged over the 10 repetitions for the different initial values of $\boldsymbol{s}_q$. We observe that the aligned variant is able to achieve small errors in $\boldsymbol{m}_q$ and $\boldsymbol{s}_q$ simultaneously while the alternatives cannot. To see how the MPAEs for $\boldsymbol{m}_q$ and $\boldsymbol{s}_q$ behave individually w.r.t. the initial value for $\boldsymbol{s}_q$ refer to Appendix H. The natural gradient as well as the aligned natural gradient method performed slightly worse than the aligned method in this experiment and we report results for them in the appendix as well.

**Longer runs do not help vanilla DPVI** Figure 1b suggests that vanilla DPVI has not converged in terms of MPAE, in the allotted number of iterations for an initial $\boldsymbol{\sigma}_q = 1$. An obvious solution then seems to be to run the inference for longer. We now fix the initialisation of $\boldsymbol{\sigma}_q$ to 1, which designates the least relative scaling of gradients at the beginning of training and thus a best-case scenario for vanilla DPVI. We vary the number of training epochs from 200 to 8000 while always keeping the privacy budget fixed at $\varepsilon = 1$. Since longer runs require more accesses to the data, the DP perturbation scale increases with the number of iterations. As before, we repeat 10 inference runs for each parameter choice.

Figure 2 shows the final MPAE over all 10 repetitions and all parameters in the Poisson regression part of the model for the different numbers of total epochs.[4] The upper panel shows that with an increasing number of epochs, the difference in MPAE of variational means between vanilla and aligned DPVI vanishes. However, the lower panel shows clearly that even in a long training regime, vanilla DPVI still does not converge in variational variances and is consistently beaten by our aligned variant.

**Iterate averaging increases robustness of downstream task** Next, we test the iterate averaging of noisy parameters traces for the generative model. We use the linear regression technique discussed in Sec. 3.1.2, individually for each parameter, to determine the length of the trace to average. We then use synthetic data from the generative model to learn the Poisson regression model used by Niedzwiedz et al. (2020) and compare the regression coefficients against those obtained from the original data. Further details on the downstream analysis setup are given in Appendix I. Figure 3 shows that the results from iterate averaged model are less noisy compared to just using the last iterate as the true parameters. However, the approach appears to be somewhat unstable: Changing the initial values of $\boldsymbol{\sigma}_q$ to 0.1 causes the variance of error for $\varepsilon = 1$. to increase over the non-averaged case. This is likely due to the simple linear regression heuristic we used in this experiment not detecting convergence correctly in this case.

---

[4]Note that it is *not* showing the evolution of the error over a single training of 8000 epochs.

### 4.2.1 Experiments on the US Census data set

To ensure the results are not specific to only a single data set, we next applied DPVI on a large set of US Census 1990 data (from UCI Dua & Graff (2017)), in the same data sharing setting as the UK Biobank experiment. We focused on individuals having military background, and using the similar mixture model/Poisson regression combination as with the UK Biobank experiment, we tried to predict the poverty indicator of the individual given military service related and demographic features. After preprocessing the data comprised of 320 754 samples with 13 features. The hyperparameters for the different DPVI variants were identical to the ones used in UK Biobank experiment.

Figure 4 shows again that the aligned variant learns the variational scales better in terms of the mean proportional absolute error (MPAE) of DPVI parameters to the non-private optimum, as was the case with UK Biobank data.

Figure 4: **US Census experiment:** Aligned DPVI method learns the variational standard deviations faster than the alternatives. The plot shows the average MPAE and the standard deviation across 10 independent repeats of the variational inference with $\boldsymbol{\sigma}_q$ initialized to 1.0 applied on the US Census data sharing experiment.

### 4.3 Logistic regression with the Adult data set

As the UK Biobank data is access-restricted, we further demonstrate our methods on the publicly available Adult data set from the UCI machine learning repository (Dua & Graff, 2017), which contains 30 162 training records. We learn a logistic regression model, classifying whether the income feature of the data exceeds $50k based on all other features.

**Aligned natural gradient outperforms the other variants** We compare our private logistic regression coefficients to the ones obtained using privacy-agnostic VI. We also test the aligned natural gradient and the natural gradient variants for this data. Figure 5a shows the $\ell_2$-norm between variational parameters learned

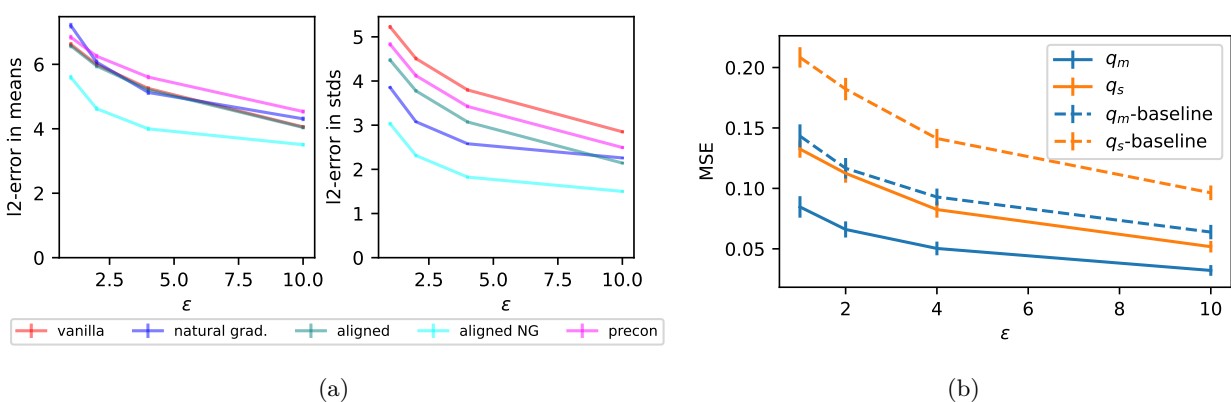

(a)                                                                 (b)

Figure 5: **Adult logistic regression experiment:** (a) The aligned natural gradient method is closest to the non-private variational parameters. Error is computed as a mean $\ell_2$-norm against non-private baseline over 20 repeats. Error bars show the standard error of the mean. (b) The standard deviation inferred from the converged traces (determined by the linear regression method) is close to that computed over last iterates across different repeats. Lines show the average MSE, error bars show the standard deviation across repeats. The baselines show the mean squared norm of the standard deviation estimated from repeated runs, corresponding to the MSE for not estimating DP-induced noise.

with and without DP. From this figure we see that the aligned natural gradient method clearly outperforms all the other variants in this setting. Additionally, we clearly see again that vanilla DPVI learns the stds poorly, and also that the natural gradient variant reverses the problem compared to vanilla and struggles in learning the variational means as suggested in Section 3.1.1.

**DP noise can be inferred from the (converged) traces** We test how well we can recover the DP noise effect from the parameter traces. We limit the test to the coefficients that have converged according to the linear regression test described in Section 3.1.2. Based on our internal tests, we chose a slope of 0.05 as the threshold for convergence. We compare the standard deviation of the converged parameter trace to an estimate of the DP-induced noise estimated by the standard deviation of the last iterates over 50 repeats in terms of mean squared error over different parameter sites.

Figure 5b shows that the noise std estimated from the converged traces is close to the noise std we have across the last iterates of multiple independent repeats.

## 4.4 Experiments with full-rank covariance

As a final experiment, we investigate aligned gradients for a full-rank Gaussian posterior approximation. We perform Bayesian linear regression over a simulated data set where we can control the number of feature dimensions as well as the strength of correlations. We control the latter by setting the rate of nonzero off-diagonal entries in the covariance matrix for simulated data points. Further details of the experimental setup can be found in Appendix J.

Figure 6 confirms that aligned gradients improve the average predictive log-likelihood of the posterior approximation over a held-out test set. This is true even when the data is not strongly correlated (panels in the right column), as the large increase in parameters over which vanilla DPVI has to split the privacy budget negatively impacts the learning.

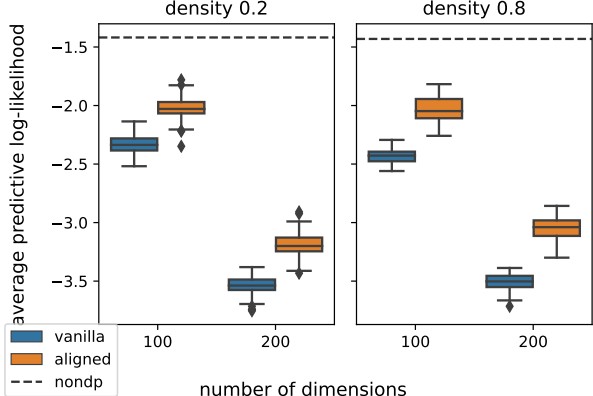

Figure 6: **Full-rank covariance experiment:** DPVI with aligned gradients achieves better predictive log-likelihood for a full-rank approximation than vanilla DPVI. Higher density means more nonzero entries in data covariance. $\varepsilon = 1$. 50 repetitions.

## 5 Discussion

In this paper we introduced the aligned gradient solution for the specific task of learning a Gaussian variational posterior. The technique should be applicable also in other tasks where gradients with respect to different parameters depend on data through a common term. Detection of such cases could be even automated by inspecting how the data enters the computational graph of the task. This would be an interesting future direction.

A limitation of DP in general is that it guarantees indistinguishability among the individuals in the data set by aiming to preserve more common characteristics of the data. Therefore the utility of a DP algorithm might be worse for individuals from less common groups.

The somewhat unexpected performance of aligned natural gradients in the UKB example might be due to bad hyperparameter choices. While we performed some hyperparameter tuning with limited success, a more comprehensive search would be needed to fully assess the performance of these methods.

The literature on MCMC holds many existing diagnostics for the converge of chains, such as the (split) R-hat estimator (Gelman & Rubin, 1992; Vehtari et al., 2021), that could be used to test the convergence of the parameter traces as well. Tuning these methods to work well in diagnosing the converge of the parameter trace would require extensive testing which we leave for future work.

## Acknowledgements

This work was supported by the Research Council of Finland (Flagship programme: Finnish Center for Artificial Intelligence, FCAI; and grants 325572, 325573), the Strategic Research Council (SRC) established within the Research Council of Finland (grant 336032), UKRI Turing AI World-Leading Researcher Fellowship, EP/W002973/1 as well as the European Union (Project 101070617). Views and opinions expressed are however those of the author(s) only and do not necessarily reflect those of the European Union or the European Commission. Neither the European Union nor the granting authority can be held responsible for them. The authors also acknowledge the computational resources provided by the Aalto Science-IT project.

This research has been conducted using the UK Biobank Resource under Application Number 65101. This work used data provided by patients and collected by the NHS as part of their care and support (Copyright © 2021, NHS England. Re-used with the permission of the NHS England and UK Biobank. All rights reserved). This work used data assets made available by National Safe Haven as part of the Data and Connectivity National Core Study, led by Health Data Research UK in partnership with the Office for National Statistics and funded by UK Research and Innovation (grant MC_PC_20058).

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

# Appendices

## A  Intuitive reasoning why larger noise would slow convergence

We consider only a single optimisation step in a single dimension for simplicity. Assume we are at $\theta^{(t)}$ and have noisy gradient

$$g^{(t)} = \nabla \mathcal{L}(\theta^{(t)}) + \eta, \eta \sim \mathcal{N}(0, \sigma^2) \tag{A.1}$$

for some perturbation scale $\sigma$. We update the parameter as

$$g^{(t+1)} = \theta^{(t)} - \alpha g^{(t)} \tag{A.2}$$

with learning rate $\alpha$.

In order to get closer to the optimum, we want $sign(g^{(t)}) = sign(\nabla \mathcal{L}(\theta^{(t)}))$. Assume wlog that $\nabla \mathcal{L}(\theta^{(t)}) > 0$, then

$$\Pr\left[sign(g^{(t)}) = sign(\nabla \mathcal{L}(\theta^{(t)}))\right] = \Pr\left[\nabla \mathcal{L}(\theta^{(t)}) + \eta > 0\right] \tag{A.3}$$

$$= \Pr\left[\eta > -\nabla \mathcal{L}(\theta^{(t)})\right] \tag{A.4}$$

$$= 1 - \Pr\left[\eta \le -\nabla \mathcal{L}(\theta^{(t)})\right] \tag{A.5}$$

$$= 1 - \Phi(-\nabla \mathcal{L}(\theta^{(t)})) \tag{A.6}$$

$$= \Phi(\nabla \mathcal{L}(\theta^{(t)})) \tag{A.7}$$

$$= \frac{1}{2}\left(1 + \mathrm{erf}\left(\frac{\nabla \mathcal{L}(\theta^{(t)})}{\sigma\sqrt{2}}\right)\right). \tag{A.8}$$

$\mathrm{erf}(\cdot)$ is a monotonically increasing function, so we see from the above that the probability of progressing towards the optimum decreases with decreasing $\frac{\nabla \mathcal{L}(\theta^{(t)})}{\sigma}$. I.e., for a fixed gradient, larger variance $\sigma^2$ will decrease the probability of progressing towards the optimum in each step.

## B  Proof of Proposition 3.1

We begin by restating Proposition 3.1.

**Proposition B.1 (Proposition 3.1).** Assume $q$ to be diagonal Gaussian, then the gradient $\boldsymbol{g}_s$ in Equation (8) becomes

$$\boldsymbol{g}_s = \boldsymbol{\eta} T'(\boldsymbol{s}_q)\boldsymbol{g}_m + \nabla_{\boldsymbol{s}_q} H(q),$$

where $T'$ denotes the derivative of $T$.

*Proof.* We first recall the reparametrisation for the diagonal Gaussian approximation from Eq. (6) as $\boldsymbol{\theta}(\boldsymbol{\eta}; \boldsymbol{m}_q, \boldsymbol{s}_q) = \boldsymbol{m}_q + T(\boldsymbol{s}_q)\boldsymbol{\eta}$ and observe that $\nabla_{\boldsymbol{m}_q}\boldsymbol{\theta} = 1$ and $\nabla_{\boldsymbol{s}_q}\boldsymbol{\theta} = \boldsymbol{\eta} T'(\boldsymbol{s}_q)$ (where we abbreviate $\boldsymbol{\theta}(\boldsymbol{\eta}; \boldsymbol{m}_q, \boldsymbol{s}_q)$ to simply $\boldsymbol{\theta}$). With this we obtain the gradient of the ELBO with respect to $\boldsymbol{m}_q$ by applying the chain rule in Eq. (7) as:

$$\boldsymbol{g}_m = \nabla_{\boldsymbol{m}_q}\mathcal{L}(q) = \nabla_{\boldsymbol{\theta}}\log p(D, \boldsymbol{\theta}) \tag{B.1}$$

Similarly applying the chain rule in Eq. 8 and inserting the above yields

$$\boldsymbol{g}_s = \nabla_{\boldsymbol{s}_q}\log p(D, \boldsymbol{\theta}) + \nabla_{\boldsymbol{s}_q} H(q) \tag{B.2}$$

$$= \nabla_{\boldsymbol{\theta}}\log p(D, \boldsymbol{\theta})\nabla_{\boldsymbol{s}_q}\boldsymbol{\theta} + \nabla_{\boldsymbol{s}_q} H(q) \tag{B.3}$$

$$= \nabla_{\boldsymbol{\theta}}\log p(D, \boldsymbol{\theta})\boldsymbol{\eta} T'(\boldsymbol{s}_q) + \nabla_{\boldsymbol{s}_q} H(q) \tag{B.4}$$

$$= \boldsymbol{\eta} T'(\boldsymbol{s}_q)\boldsymbol{g}_m + \nabla_{\boldsymbol{s}_q} H(q). \tag{B.5}$$

$\square$

## C  Proof of $T'(s_q) \leq T(s_q)$ for softplus and exponential function

$T(s_q) = \mathbf{softplus}(s_q)$   Consider we transform $s_q$ in positive real numbers using the softplus function:

$$T(s_q) = \log(1 + \exp(s_q)). \tag{C.1}$$

First, we make the following observation which connects the softplus to the sigmoid function

$$
\begin{aligned}
T(s_q) &= -\log\left(\frac{1}{1 + \exp(s_q)}\right) \\
&= -\log\left(1 - \frac{1}{1 + \exp(-s_q)}\right) \\
&= -\log\left(1 - \sigma(s_q)\right),
\end{aligned}
\tag{C.2}
$$

where $\sigma$ denotes the sigmoid function. We then get $T'(s_q) = \sigma(s_q)$. It is easy to see that $\log(x) \leq x - 1$ and hence

$$
\begin{aligned}
T(s_q) &= -\log\left(1 - \sigma(s_q)\right) \\
&\geq 1 - (1 - \sigma(s_q)) = \sigma(s_q) = T'(s_q).
\end{aligned}
\tag{C.3}
$$

We have therefore shown that $T(s_q) \geq T'(s_q) \, \forall s_q \in \mathbb{R}$.

$T(s_q) = \exp(s_q)$   For $T(s_q) = \exp(s_q)$ the proof follows immediately from the fact that $T'(s_q) = \exp(s_q) = T(s_q)$.

## D  Proof of Theorem 3.1: Variance in aligned scale gradients is smaller

We begin by restating the theorem:

**Theorem D.1** (**Theorem 3.1**). Assume that $C^{vanilla} \geq C^{aligned}$. If we obtain $\boldsymbol{\sigma}_q$ through transformation $T$ such that $T'(s) \leq 1$, then for any fixed batch,

$$\mathrm{Var}_{\boldsymbol{\eta}, \boldsymbol{\psi}}\left[\tilde{\boldsymbol{g}}_s^{aligned}\right] < \mathrm{Var}_{\boldsymbol{\eta}, \boldsymbol{\psi}}\left[\tilde{\boldsymbol{g}}_s^{vanilla}\right],$$

where $\boldsymbol{\eta} \in \mathcal{N}(0, 1)$ is the random variable of the MC approximation to the ELBO in the reparametrisation approach and $\boldsymbol{\psi} \in \mathcal{N}(0, 1)$ that of the DP perturbation.

*Proof.* We want to show, that the variance of the perturbed std parameter gradient for the aligned method is less or equal than the one for the vanilla approach in all the dimensions of the gradient. We start by setting the clipping threshold for the vanilla approach as $C$ and the one for the aligned as $C_a$. Denote the $j$th dimension of $\boldsymbol{g}_s$ obtained from the vanilla approach as $\tilde{\boldsymbol{g}}_{s,j}$, and similarly $\tilde{\boldsymbol{g}}_{s,j}^{aligned}$ for the aligned. For the vanilla approach, we have

$$\mathrm{Var}_{\boldsymbol{\eta}, \boldsymbol{\psi}}\left[\tilde{\boldsymbol{g}}_{s,j}\right] = \mathrm{Var}_{\boldsymbol{\eta}, \boldsymbol{\psi}}\left[\boldsymbol{\eta}_j T'(s_{q,j})\frac{\partial}{\partial \boldsymbol{m}_{q,j}}\mathcal{L}(q) + \frac{\partial}{\partial s_{q,j}}H(q) + \boldsymbol{\psi}_j \sigma_{DP} C\right] \tag{D.1}$$

$$= \mathrm{Var}_{\boldsymbol{\eta}}\left[\boldsymbol{\eta}_j T'(s_{q,j})\frac{\partial}{\partial \boldsymbol{m}_{q,j}}\mathcal{L}(q)\right] + \sigma_{DP}^2 C^2 \, \mathrm{Var}_{\boldsymbol{\psi}}\left[\boldsymbol{\psi}_j\right] \tag{D.2}$$

$$= \mathrm{Var}_{\boldsymbol{\eta}}\left[\boldsymbol{\eta}_j T'(s_{q,j})\frac{\partial}{\partial \boldsymbol{m}_{q,j}}\mathcal{L}(q)\right] + \sigma_{DP}^2 C^2, \tag{D.3}$$

and for the aligned

$$\text{Var}_{\boldsymbol{\eta},\boldsymbol{\psi}}\left[\tilde{\boldsymbol{g}}_{s,j}^{aligned}\right] = \text{Var}_{\boldsymbol{\eta},\boldsymbol{\psi}}\left[\boldsymbol{\eta}_j T'(\boldsymbol{s}_{q,j})\frac{\partial}{\partial \boldsymbol{m}_{q,j}}\mathcal{L}(q) + \frac{\partial}{\partial \boldsymbol{s}_{q,j}}H(q) + \boldsymbol{\eta}_j T'(\boldsymbol{s}_{q,j})\boldsymbol{\psi}_j\sigma_{DP}C_a\right] \tag{D.4}$$

$$= \text{Var}_{\boldsymbol{\eta}}\left[\mathbb{E}_{\boldsymbol{\psi}|\boldsymbol{\eta}}\left[\boldsymbol{\eta}_j T'(\boldsymbol{s}_{q,j})\frac{\partial}{\partial \boldsymbol{m}_{q,j}}\mathcal{L}(q) + \frac{\partial}{\partial \boldsymbol{s}_{q,j}}H(q) + \boldsymbol{\eta}_j T'(\boldsymbol{s}_{q,j})\boldsymbol{\psi}_j\sigma_{DP}C_a\right]\right] \tag{D.5}$$

$$+ \mathbb{E}_{\boldsymbol{\eta}}\left[\text{Var}_{\boldsymbol{\psi}|\boldsymbol{\eta}}\left[\boldsymbol{\eta}_j T'(\boldsymbol{s}_{q,j})\frac{\partial}{\partial \boldsymbol{m}_{q,j}}\mathcal{L}(q) + \frac{\partial}{\partial \boldsymbol{s}_{q,j}}H(q) + \boldsymbol{\eta}_j T'(\boldsymbol{s}_{q,j})\boldsymbol{\psi}_j\sigma_{DP}C_a\right]\right] \tag{D.6}$$

$$= \text{Var}_{\boldsymbol{\eta}}\left[\boldsymbol{\eta}_j T'(\boldsymbol{s}_{q,j})\frac{\partial}{\partial \boldsymbol{m}_{q,j}}\mathcal{L}(q)\right] + \mathbb{E}_{\boldsymbol{\eta}}\left[\boldsymbol{\eta}_j^2 T'(\boldsymbol{s}_{q,j})^2\sigma_{DP}^2 C_a^2\right] \tag{D.7}$$

$$= \text{Var}_{\boldsymbol{\eta}}\left[\boldsymbol{\eta}_j T'(\boldsymbol{s}_{q,j})\frac{\partial}{\partial \boldsymbol{m}_{q,j}}\mathcal{L}(q)\right] + T'(\boldsymbol{s}_{q,j})^2\sigma_{DP}^2 C_a^2. \tag{D.8}$$

We can easily see that the two variances differ only in the noise term. Now, it is easy to see, that if we set $C_a = C$ and have a transformation $T$ s.t. $T'(s) \leq 1$, we have

$$\text{Var}_{\boldsymbol{\eta},\boldsymbol{\psi}}\left[\tilde{\boldsymbol{g}}_{s,j}^{aligned}\right] \leq \text{Var}_{\boldsymbol{\eta},\boldsymbol{\psi}}\left[\tilde{\boldsymbol{g}}_{s,j}\right], \forall j. \tag{D.9}$$

Setting $C = C_a$ already shows that the aligned method's variance cannot exceed the vanilla method. However, consider now that $C_a$ is a clipping threshold that satisfies $\text{Pr}(||\boldsymbol{g}_m|| > C_a) < \alpha$ for some $\alpha \in [0, 1)$, i.e. only an $\alpha$ fraction of the gradients get clipped in the aligned approach. Now, since the vanilla method clips based on the norm $||\boldsymbol{g}||^2 = ||\boldsymbol{g}_m||^2 + ||\boldsymbol{g}_s||^2 \geq ||\boldsymbol{g}_m||^2$, we need to set the $C$ at least as large as $C_a$ to facilitate the same probability of clipping. Furthermore, we have the equality $||\boldsymbol{g}||^2 = ||\boldsymbol{g}_m||^2$ only in the case when $||\boldsymbol{g}_s|| = 0$ which would only happen if the std parameter has converged. Therefore, the $C$ should be chosen larger than $C_a$ to avoid clipping prior to the convergence of $\boldsymbol{s}_q$. Note that the magnitude of the $\boldsymbol{g}_s$ does not affect the aligned approach, and therefore the clipping bound $C_a$. Now, as we need to choose $C > C_a$ and we have assumed $T'(s) \leq 1$, we have

$$\text{Var}_{\boldsymbol{\eta},\boldsymbol{\psi}}\left[\tilde{\boldsymbol{g}}_{s,j}^{aligned}\right] < \text{Var}_{\boldsymbol{\eta},\boldsymbol{\psi}}\left[\tilde{\boldsymbol{g}}_{s,j}\right], \forall j. \tag{D.10}$$

Furthermore, from the above variance expressions, it is easy to see that the variance in the aligned approach gets significantly smaller than the vanilla one when $T'(s) \ll 1$. This would be the case for example if we initialize the variational posterior with small $\boldsymbol{s}_q$, or similarly towards the end of the convergence for models with small posterior variance. $\square$

## E  Variance of DP from OU process on convergence

We follow closely the analysis performed by Mandt et al. (2017), Sec. 3.2., which makes the following assumptions for the loss function $\mathcal{L}$ around its optimum $\boldsymbol{\xi}^*$:

1. Mini-batch gradients of the loss functions are well approximated by a zero-mean Gaussian distribution with covariance matrix $\frac{1}{S}\boldsymbol{Z}$, where $S$ denotes the size of mini-batch,

2. $\mathcal{L}$ is locally well approximated by a quadratic function.

We make the additional assumption that the clipping threshold $C$ is chosen such that no clipping occurs for gradients close to the optimum in order to avoid clipping-induced bias.

We begin by stating the SGD parameter update equation and the resulting update step $\Delta\boldsymbol{\xi}(t)$ for update steps close to the optimum.

$$\boldsymbol{\xi}(t+1) = \boldsymbol{\xi}(t) - \alpha \left( \nabla_{\boldsymbol{\xi}} \mathcal{L}(\boldsymbol{\xi}(t)) + \frac{1}{\sqrt{S}} \boldsymbol{B}\boldsymbol{\nu} + \sigma_{DP} \boldsymbol{I}\boldsymbol{\psi} \right) \ , \ \boldsymbol{\nu} \sim \mathcal{N}(\boldsymbol{0}, \boldsymbol{I}) \ , \ \boldsymbol{\psi} \sim \mathcal{N}(\boldsymbol{0}, \boldsymbol{I})$$

$$= \boldsymbol{\xi}(t) - \alpha \left( \nabla_{\boldsymbol{\xi}} \mathcal{L}(\boldsymbol{\xi}(t)) + \left( \frac{1}{\sqrt{S}} \boldsymbol{B} + \sigma_{DP} \boldsymbol{I} \right) \boldsymbol{\psi}' \right) \ , \ \boldsymbol{\psi}' \sim \mathcal{N}(0, \boldsymbol{I}) \tag{E.1}$$

$$\Delta\boldsymbol{\xi}(t) := \boldsymbol{\xi}(t+1) - \boldsymbol{\xi}(t) = -\alpha \nabla_{\boldsymbol{\xi}} \mathcal{L}(\boldsymbol{\xi}(t)) - \alpha \left( \frac{1}{\sqrt{S}} \boldsymbol{B} + \sigma_{DP} \boldsymbol{I} \right) \boldsymbol{\psi}' \tag{E.2}$$

We denote with $\boldsymbol{B}$ the triangular matrix resulting from Cholesky decomposition of $\boldsymbol{Z} = \boldsymbol{B}\boldsymbol{B}^T$. Since we are performing DP-SGD, we have an additional independent noise term with scale $\sigma_{DP}$. However, as both sources of stochasticity are independent zero-mean Gaussians, we can easily reformulate using a single Gaussian source of noise with the total variance.

We now restate E.2 by the following stochastic differential equation:

$$d\boldsymbol{\xi}(t) = -\alpha \nabla_{\xi} \mathcal{L}(\boldsymbol{\xi}(t)) - \alpha(\frac{1}{\sqrt{S}} \boldsymbol{B} + \sigma_{DP} \boldsymbol{I}) d\boldsymbol{W}(t) \tag{E.3}$$

From our second assumption, we know that

$$\mathcal{L}(\boldsymbol{\xi}) \approx \frac{1}{2}(\boldsymbol{\xi} - \boldsymbol{\xi}^*)^T \boldsymbol{A}(\boldsymbol{\xi} - \boldsymbol{\xi}^*), \tag{E.4}$$

where $\boldsymbol{A} = \frac{\partial^2}{(\partial \boldsymbol{\xi})^2} \mathcal{L}(\boldsymbol{\xi}^*)$.

Inserting (E.3) in (E.4), we get

$$d\boldsymbol{\xi}(t) = -\alpha \boldsymbol{A}(\boldsymbol{\xi} - \boldsymbol{\xi}^*)dt + \alpha(\frac{1}{\sqrt{S}} \boldsymbol{B} + \sigma_{DP} \boldsymbol{I}) dW(t) \tag{E.5}$$

with describes an Ornstein-Uhlenbeck (OU) process with Gaussian stationary distribution

$$q(\boldsymbol{\xi}) \propto \exp \left\{ -\frac{1}{2}(\boldsymbol{\xi} - \boldsymbol{\xi}^*)^T \boldsymbol{\Sigma}^{-1}(\boldsymbol{\xi} - \boldsymbol{\xi}^*) \right\} \tag{E.6}$$

where $\Sigma$ satisfies the Lyapunov equation

$$\boldsymbol{\Sigma}\boldsymbol{A} + \boldsymbol{A}\boldsymbol{\Sigma} = \alpha \left( \frac{1}{S}\boldsymbol{Z} + \sigma_{DP}^2 \boldsymbol{I} \right). \tag{E.7}$$

Even without determining $\boldsymbol{A}$ we can already make an important discovery from this: Since $\boldsymbol{A}$ is fixed around the optimum, we see that the noise covariance of our OU process scales linearly with respect to $\sigma_{DP}^2$.

Note that the above analysis holds for a constant learning rate. We have used the Adam optimization method (Kingma & Ba, 2015) in our experiments throughout the paper. While Adam does adapt the learning rate, recently Mohapatra et al. (2021) showed that the learning rate of Adam will converge to a static value, which means that the analysis above still holds as it is only concerned with the learning rate at convergence.

## F  Hyperparameters

We use Adam (Kingma & Ba, 2015) as the optimiser for all the experiments with starting learning rate of $10^{-3}$. In all of our experiments, the $\delta$ privacy parameter was set to $1/N$ where $N$ denotes the size of the training data.

**For the UKB experiment**  In the experiments, we used various different training lengths (depicted e.g. in Figure 2). For all of our runs, we set the subsampling rate as 0.01. The clipping threshold $C$ was set to $C = 2.0$ for the aligned and vanilla, 4.0 for the preconditioned variant and to 0.1 for the natural gradient based variants.

**For the Adult experiment** The training was run for $4\,000$ epochs with subsampling ratio of $0.01$, corresponding to total of $400\,000$ gradient steps.

We chose the clipping thresholds for the gradient perturbation algorithm as the 97.5% upper quantile of the training data gradient norms at the non-private optima. This was done to avoid clipping-induced bias, thus making the models comparable to the non-private baseline. This lead to clipping thresholds $C$ presented in Table F.1.

Table F.1: Clipping thresholds for the Adult data logistic regression model

| Variant | C |
|---|---|
| Aligned | 3.0 |
| Aligned Natural Grad. | 0.1 |
| Natural Grad. | 0.1 |
| Vanilla | 3.0 |
| Preconditioned | 4.0 |

## G  Model priors

### G.1  For the UKB and the US Census experiments

Recall the probabilistic model used in the experiments:

$$p(\boldsymbol{X} \mid \boldsymbol{\theta_X}, \boldsymbol{\pi}) = \sum_{k=1}^{K} \pi_k \prod_{j=1}^{d} \mathrm{Categorical}(\boldsymbol{X}_j \mid \boldsymbol{\theta}_{\boldsymbol{X}}^{(k)}) \tag{G.1}$$

$$p(\boldsymbol{y} \mid \boldsymbol{X}, \boldsymbol{\theta_y}) = \mathrm{Poisson}(\boldsymbol{y} \mid \exp(\boldsymbol{X}\boldsymbol{\theta_y})). \tag{G.2}$$

The categorical probabilities $\boldsymbol{\theta}_{\boldsymbol{X}_j}^{(k)}$ for each of the categorical features $\boldsymbol{X}_j$, were given a uniform Dirichlet$(\mathbf{1})$ prior. Similarly the mixture weights $\boldsymbol{\pi}$ we assigned a uniform Dirichlet prior. The regression coefficients $\boldsymbol{\theta_y}$ were given a std. normal $N(0, I)$ prior.

### G.2  For the Adult experiment

We use the following model and prior

$$\boldsymbol{y} \sim \sigma(\boldsymbol{X}\boldsymbol{w}), \tag{G.3}$$

$$\boldsymbol{w} \sim \mathcal{N}(\mathbf{0}, \boldsymbol{I}), \tag{G.4}$$

where $\sigma(\cdot)$ denotes the logistic regression function, $\sigma(x) = 1/1+\exp\{-x\}$.

## H  More results for robustness

Figure H.1 shows how the MPAE for the different DPVI variants behave for the different initial values for $\boldsymbol{s}_q$ for both, variational mean (upper panel) and standard deviation (lower panel) after $1\,000$ epochs.

Figure H.2 shows the parameters traces for $\boldsymbol{s}_q$ initialised such that $\boldsymbol{\sigma}_q = 0.01$ (left) and $\boldsymbol{\sigma}_q = 0.1$ (right) with the same split in upper and lower panels, similar to Figure 1b for $\boldsymbol{\sigma}_q = 1.0$ in the main body of the paper.

We observe that with decreasing initial values for $\boldsymbol{s}_q$ $(/\boldsymbol{\sigma}_q)$ it becomes increasingly difficult for vanilla DPVI to learn variational standard deviation but learning of means is slightly improved. Preconditioned DPVI performs better overall in terms of standard deviation but learns means worse. Aligned DPVI consistently outperforms both competing variants.

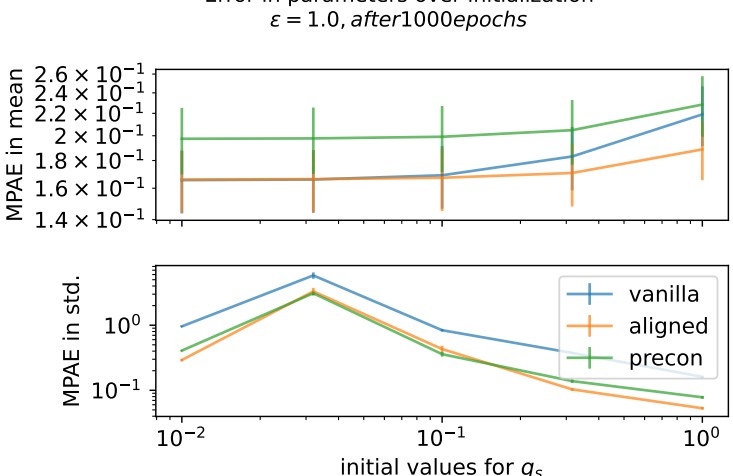

Figure H.1: The aligned variant has consistently low error across different initial values of $q_\sigma$. Figure shows the error for $\epsilon = 1$ for 1000 epochs of training.

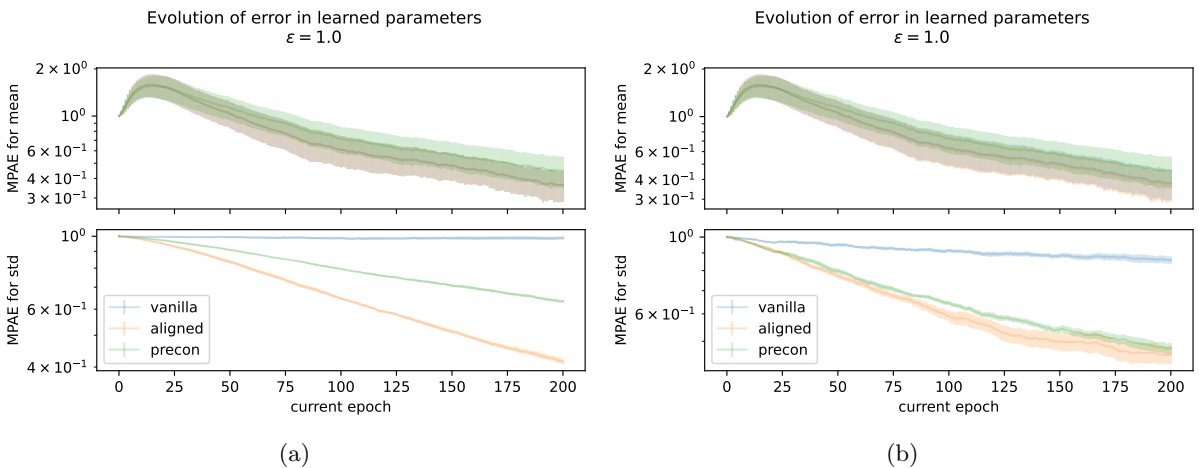

Figure H.2: Aligned DPVI consistently converges faster than the other methods for different initialisation of $\boldsymbol{s}_q$. **On left**, the $\boldsymbol{s}_q$ is initialised such that $\boldsymbol{\sigma}_q = 0.01$ and **on right** such that $\boldsymbol{\sigma}_q = 0.1$.

## H.1 Natural gradients and the aligned natural gradients in the UKB experiment

Besides the vanilla and aligned variant, we also fitted the UKB model using the natural gradient and aligned natural gradient variants. From Figure H.3 we can see the trade-off natural gradients make; the means are learned worse than standard deviations, which is what we expect based on the analysis of Section 3.1.1. Somewhat surprisingly, the aligned natural grad. variant performs slightly worse than the aligned variant in this experiment. This might be due to poor choice of hyperparameter, for example the learning rate for the Adam optimiser used in the experiments was set to $10^{-3}$ for all the variants, while we know that the natural grad. variants tend to have smaller gradients than the others - although Adam should in theory be able to adapt to that.

To evaluate whether the difference between aligned and aligned natural grad. results is due to clipping threshold being too large or too small for the aligned natural grad. approach, thus perturbing the gradients excessively or introducing excessive clipping bias, we repeat the experiment with higher and lower clipping thresholds. Figure H.4a shows that increasing the clipping threshold to $C = 0.1$ harms the natural gradient

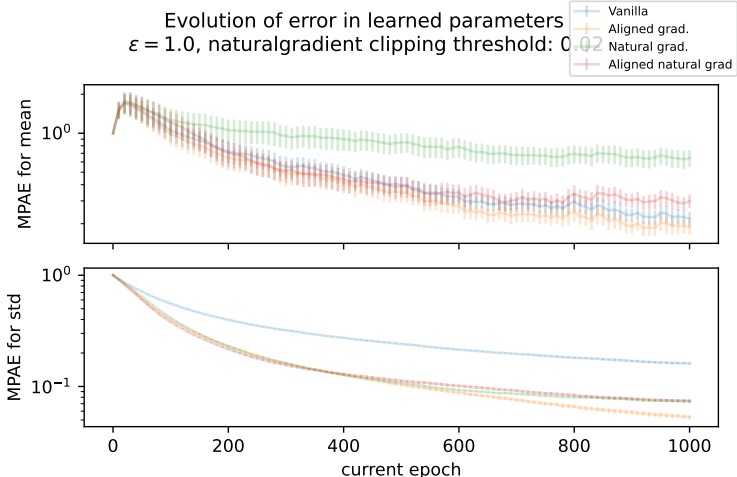

Figure H.3: The natural gradient based variants variant achieve comparable performance in terms of variational parameters for std but perform worse for variational means. This is strongly observable for the plain natural gradients while aligned natural gradients improve error in variational means in comparison. However, both are outperformed by (non-natural) aligned gradients in this experiment. Lines show the mean MPAE over 10 independent repeats as well as std. of mean as error bars. Clipping threshold for both natural gradient based variants is set to $C = 0.02$, $\boldsymbol{\sigma}_q$ was initialised to 1. The natural gradient variant appears to diverge from the true variational std. which might be caused by clipping-induced bias, while still struggling to learn the correct variational mean.

based variants, most likely due to introducing too much privacy noise and thus preventing convergence of the variational mean parameters. Conversely, Figure H.4b shows that setting clipping threshold $C = 0.01$ natural gradient variants is too small, and the aligned natural gradients start to suffer from clipping-induced bias.

These observations highlight the importance of choosing the clipping threshold appropriately to the method, which is not a trivial task. For this experiment, this seems to have a direct influence on how well aligned natural gradients can reign in the inversal of relative scaling of components present in plain natural gradients. For future work, it would be interesting to perform a more extensive hyperparameter tuning including for example the learning rate and number of iterations as the tunable hyperparameters, to see if the natural gradient methods make some non-trivial trade-offs in the learning that differ fundamentally from the aligned and vanilla methods.

# I   Further details on downstream analysis for UKB data

In the UKB experiment, we use the learned variational posterior to sample a synthetic data set from the posterior predictive distribution (PPD) as suggested by Jälkö et al. (2021). We test the method by comparing the synthetic data in downstream analysis to the original data. As the downstream task, we fitted a Poisson regression model that aims to predict whether individual catches SARS-CoV-2 based on the predictors in the data. Note that this downstream perfectly overlaps with our generative model.

In order to properly reflect the uncertainty rising from the data generating process to the final results computed from the synthetic data, we will employ so called *Rubin's rules* (Rubin, 2004). In this procedure, we first sample multiple synthetic data sets from the PPD and compute the downstream analysis on each of the sampled synthetic data. Next, the results are aggregated according to a set of rules and we recover finally a more robust estimator for our downstream analysis. Further discussion about the Rubin's rules can be found for example in (Reiter & Raghunathan, 2007).

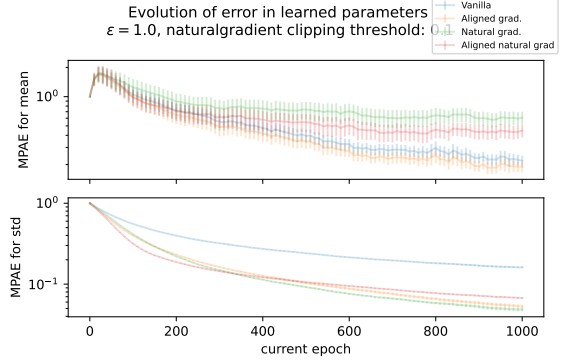
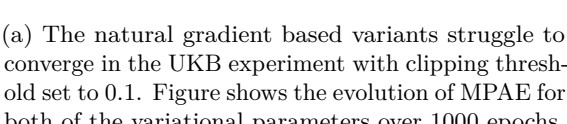
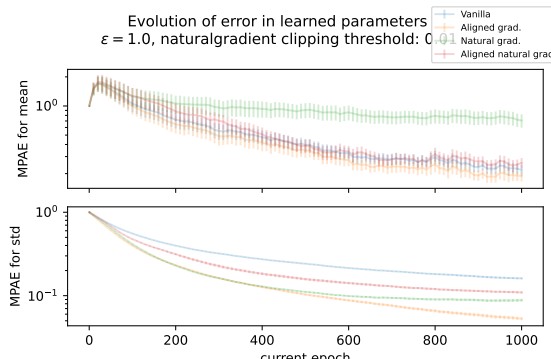

(a) The natural gradient based variants struggle to converge in the UKB experiment with clipping threshold set to 0.1. Figure shows the evolution of MPAE for both of the variational parameters over 1000 epochs.

(b) Both natural gradient variants start to diverge if clipping is set too low $C = 0.01$.

Figure H.4: Tests with smaller clipping threshold for the natural gradient variants. The clipping threshold for the vanilla and aligned is still set to 2.0.

In our experiments, we sampled 100 data sets from the PPD learned using the aligned variant, and applied the Rubin's rules to compute a mean and std. estimate for the Poisson regression coefficients. Finally, the obtained means were compared to the Poisson regression coefficients learned using the original data.

## J   Experimental setup for full-rank Gaussian approximation

In this experiment we create simulated data where we control the amount of correlations between data dimensions as the ratio $\rho$ of non-zero off-diagonal entries in the correlation matrix. To generate data with $d$ dimensions and correlation density $\rho$, we

1. generate a correlation matrix $\boldsymbol{C}$ using Algorithm J.1 with inputs $d, \rho, \alpha = 8, \beta = 10$,

2. sample a diagonal matrix $\boldsymbol{D}$ of marginal variances, where $\{D\}_{ii} \sim \exp\{\mathcal{N}(0, 0.2^2)\}$,

3. obtain the covariance matrix $\boldsymbol{\Sigma} = \boldsymbol{DCD}$

4. sample $N = 10\,000$ data points $\boldsymbol{x}_n \sim \mathcal{N}(\boldsymbol{0}, \boldsymbol{\Sigma})$

5. sample random regression weight vector $\boldsymbol{w} \sim \mathcal{N}(\boldsymbol{0}, \boldsymbol{I})$

6. sample $\boldsymbol{y} \sim \mathcal{N}(\boldsymbol{Xw}, \sigma_y^2)$, with $\sigma_y = 1$.

We perform the above for all combinations of $d = 100, 200$ and $\rho = 0.2, 0.8$. We then use vanilla DPVI and DPVI with aligned gradients to learn the full-rank Gaussian posterior approximation to the Bayesian linear regression model with priors

$$\boldsymbol{y} \sim \mathcal{N}(\boldsymbol{Xw}, \sigma_y^2), \tag{J.1}$$

$$\boldsymbol{w} \sim \mathcal{N}(\boldsymbol{0}, \boldsymbol{I}) \tag{J.2}$$

$$\sigma_y \sim \mathrm{Gamma}(0.1, 0.1). \tag{J.3}$$

We run the inference for $1\,000$ epochs, gradient clipping threshold 0.2 and subsampling ratio 0.01.

For the same $d, \rho$ we then generate another $10\,000$ data points and compute the log-likelihood using the obtained posterior approximation. We repeat the inference and evaluation 50 times for each method and combination of $d$ and $\rho$, keeping the generated training and testing set fixed.

---

**Algorithm J.1** Routine to generate a $d$-dimensional correlation matrix with given density $\rho$ and strength of correlations controlled by $\alpha, \beta$.

---

**Require:** $d, \rho \in [0,1], \alpha > 0, \beta > 0$
**Ensure:** correlation matrix $C$

  $K \leftarrow \rho \frac{d(d-1)}{2}$                                                      ▷ number of non-zero off-diagonal entries
  $U \leftarrow \emptyset$
  $k \leftarrow 0$
  $C \leftarrow I_d$
  **for** $k = 1, \ldots, K$ **do**
      sample $(i,j) \in T \setminus U$ at random
      sample $c \sim \text{Beta}(\alpha, \beta)$                    ▷ sample correlation strength, controlled by $\alpha$ and $\beta$
      sample $f \in \{-1, 1\}$ at random                               ▷ sample sign of correlation
      $C_{ij} \leftarrow fc$
      $C_{ji} \leftarrow C_{ij}$
      $U \leftarrow U \cup \{(i,j), (j,i)\}$
  **end for**

---

Table K.1: Estimated runtimes for UKB experiment

| #epochs | single repeat runtime | repeats | # epsilon values | # initial values |
|---------|----------------------|---------|------------------|------------------|
| 200 | 4-8min | 10 | 4 | 5 |
| 400 | 8-16min | 10 | 4 | 1 |
| 600 | 12-18min | 10 | 4 | 1 |
| 800 | 16-32min | 10 | 4 | 1 |
| 1000 | 20-40min | 10 | 4 | 1 |
| 2000 | 40-80min | 10 | 4 | 1 |
| 4000 | 80-160min | 10 | 4 | 1 |
| 8000 | 160-320min | 10 | 4 | 1 |

## K   Runtimes

### K.1   UKB experiments

In this experiment, we ran all the variants separately for 10 seeds and 4 levels of privacy. Additionally, we experimented with different runtimes and initialisations. For a training of 1 000 epochs, a single repeat takes between 20 to 40 minutes. The runtime scales linearly with the number of epochs.

Further, we computed the downstream task for the aligned variants, which includes generating the 100 synthetic data sets and fitting the downstream Poisson regression model on those 100 synthetic data. This procedure takes between 30 to 60 minutes to complete.

A rough estimate of the runtimes for the UKB experiment is given in Table K.1. A single CPU core with 8gb of memory was used for all the runs.

### K.2   Adult experiments

A single training repeat of learning the logistic regression model for all the different variants of DPVI, took between 10 and 30 minutes to finish on a single CPU core with 8gb of memory assigned. In total, the Adult experiment was repeated 50 times for four different levels of privacy. Therefore the total runtime of all the experiments is **between 2 000 and 6 000 minutes**.

The variance in running times is likely due to differences in computation nodes in the clustered assigned by a automatic run scheduler.

### K.3 Full-rank experiments

A single run for this experiment consisted of the inference using both vanilla and aligned DPVI with full-rank approximations. All runs were executed on a computing cluster utilising Nvidia K80, A100, P100, V100 GPU hardware, to which the runs were allocated automatically to balance overall load. As a result, runtimes varied slightly: Runs for and 100 dimensional data took 6-8 minutes to finish, runs for 200 dimensions took 8-10 minutes. With a total of 4 data set configurations and 50 repeats for each, the total runtime is 1 400 to 1 800 minutes.

## L Gradient distributions for different variants

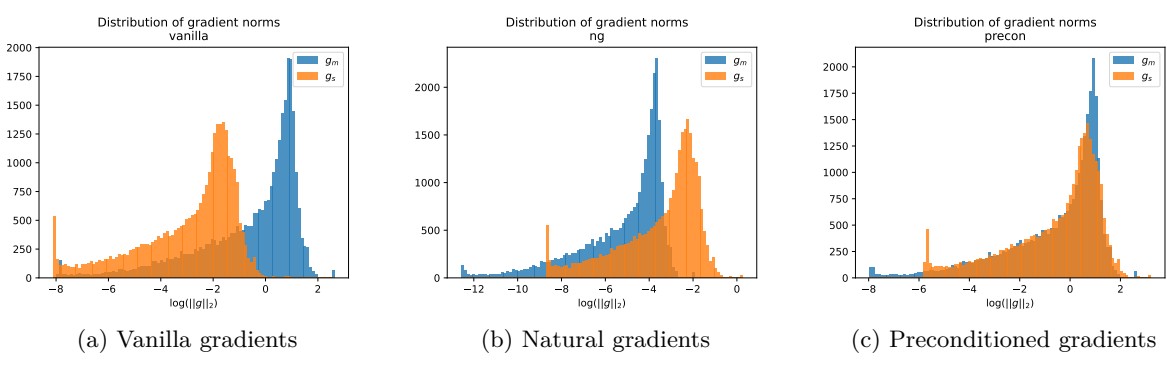

(a) Vanilla gradients      (b) Natural gradients      (c) Preconditioned gradients

Figure L.1

Figure L.1 shows the distributions of gradient norms for variational means and scales for different variants of DPVI discussed in Section 3.1.1 when $s_q$ is set to 0.1. Figure L.1a clearly shows the different magnitudes for variational standard deviation in vanilla DPVI. Figure L.1b demonstrates that natural gradients simply reverse the problem. Figure L.1c shows that the scaling approach achieves matching magnitudes quite well. However, it comes at the cost of increasing the norm of the full (combined) gradient and therefore increased sensitivity.

