# OpenReview forum: "DPVIm: Differentially Private Variational Inference Improved"
_TMLR — Accepted by TMLR_

### Review · Reviewer_n9qi · 2023-05-19

**Summary Of Contributions:**

The paper studied differentially private variational inference problems solved by gradient-based methods. A new method called the aligned gradient procedure is proposed, which provides a less noisy gradient estimator. The method can be extended to learn the full-rank covariance approximation. Experiment results in simulated data and real data are provided.

**Audience:**

Yes

**Claims And Evidence:**

Yes

**Requested Changes:**

1. The related statements about $\eta$ mentioned in Weakness 3 should be revised if I am correct.

2. More discussion on the assumption on $C$ in Thm 3.1

3. More details for experiments, i.e., how to choose $C$.

**Strengths And Weaknesses:**

Strength:
1. The message is clear and simple.
2. Experiments show that the proposed method is effective and efficiency.

Weakness:
1. The idea of Alg 1 is natural, and there is no theoretical innovation.
2. My main concern is the assumption ‘Assume C is chosen such that the bias induced by clipping is the same in vanilla and aligned DPVI’ in Thm 3.1. Is this assumption fair/reasonable? Are there any references? In addition, in the experiments, how to choose C and C’ for two methods?
3. Some statements are not rigorous. For example, In the paragraph above Eq(9), the author stated that “As $\eta\sim N(0,I_d)$, we have $\eta_j=O(1)$”. It is not correct? We could only say that with high probability, $\eta_j\le M$ where $M$ is a constant.

---

> ### Author Response · Authors · 2023-06-09
> **Response to Reviewer n9qi**
>
> **“How to choose C and C’ for two methods”**: For our experiments we chose the clipping thresholds for the gradient perturbation algorithm as the $97.5\%$ upper quantile of the training data gradient norms at the non-private optima. This is detailed in Appendix F (“Hyperparameters”). This is done to ensure a fair comparison by ensuring that clipping occurs only rarely and therefore clipping-induced bias at convergence is low for all methods. Note that practical applications would need to use a DP method to tune the clipping hyperparameter for the actual algorithm in question. However, as shown in the proof of our Theorem 3.1, in order to make the vanilla approach clip as little as the aligned approach, we do end up with larger variance for the vanilla gradients.
>
> Please also see our response to all reviewers for your remaining questions, which have been raised by other reviewers as well in a similar form.

---

### Review · Reviewer_E9ZR · 2023-05-20

**Summary Of Contributions:**

The paper considers differentially private (DP) variational inference (VI). The paper identifies a problem of "misaligned'' gradients that occurs when the standard (DP-SGD-based) solution is used: the gradients of different components of the loss functions may have different magnitudes and sensitivities, requiring different amounts of privacy noise. To address this issue, the paper proposes an "aligned gradient procedure" which reduces the variance of the estimates of the gradients of the VI loss. Moreover, the paper proposes using iterate averaging to reduce the variance of the parameter estimate. The efficacy of their methods is evaluated empirically.

**Audience:**

Yes

**Broader Impact Concerns:**

None.

**Claims And Evidence:**

No

**Requested Changes:**

Please see "**Weaknesses**'' above.

**Strengths And Weaknesses:**

**Strengths:**

-The Algorithms 1 and 2 make sense and are intuitive ways to address the misaligned gradients problem.

-Empirically, the algorithms show improvements over vanilla and preconditioned DPVI

**Weaknesses:**

-The writing is unclear in many places, with long sentences and/or insufficient explanation of the core ideas and/or excessive detail too early. Specifically:

--The Abstract is too detailed and not accessible. I am a DP expert, and was confused after reading the abstract, as there was a lot of non-standard terminology (e.g. "parameter traces'') used before it was defined. Now imagine someone reading the abstract who is only somewhat familiar with DP and/or VI. I would suggest that you re-write the abstract to focus on: explaining the problem of DP VI clearly; then explain what the previous approaches to tackling DP VI are; then explain the shortcomings of these prior approaches; then explain (at a high level, not in great detail, and without using undefined terminology) how you address these shortcomings.

--Intro:  "which is typically ignored or unknown'' is unclear. I would suggest deleting this and replacing with a new sentence: ``However, the issue of varying sensitivities is often neglected or overlooked.''

--last sentence of paragraph 1 is unclear. Delete or re-word.

--The problem of DP VI should be defined early in the introduction, *before* the particular problem of perturbed gradients in DP VI (paragraph 2) is discussed.

--``DPVI is a widely applicable...based on DP-SGD'' sentence is out of place. I suggest moving it to the second sentence of paragraph 2.

(More writing issues may be brought up later.)

-Missed references:

--While not directly relevant, it would be good to cite (e.g. in a sentence at the end of the first paragraph of the Related Work section) works that study the use of gradient clipping for obtaining optimal (at least in reference 2) below) rates in DP heavy-tailed stochastic optimization:

1) Kamath, G., Liu, X. and Zhang, H., 2022, June. Improved rates for differentially private stochastic convex optimization with heavy-tailed data. In International Conference on Machine Learning (pp. 10633-10660). PMLR.

2) Lowy, A. and Razaviyayn, M., 2023, February. Private Stochastic Optimization with Large Worst-Case Lipschitz Parameter: Optimal Rates for (Non-Smooth) Convex Losses and Extension to Non-Convex Losses. In International Conference on Algorithmic Learning Theory (pp. 986-1054). PMLR.

--When discussing iterate averaging to reduce noise in parameter estimation, it should be noted that this general idea is not novel and has been used to obtain optimal rates for various problems in DP optimization, e.g.:

1) Bassily, R., Smith, A. and Thakurta, A., 2014, October. Private empirical risk minimization: Efficient algorithms and tight error bounds. In 2014 IEEE 55th annual symposium on foundations of computer science (pp. 464-473). IEEE.

2) Bassily, R., Feldman, V., Talwar, K. and Guha Thakurta, A., 2019. Private stochastic convex optimization with optimal rates. Advances in neural information processing systems, 32.

3) Lowy, A. and Razaviyayn, M., 2021. Private federated learning without a trusted server: Optimal algorithms for convex losses. arXiv preprint arXiv:2106.09779.

-Section 2.2: explanation/motivation/intuition of ELBO (and why it is called ELBOW) would be great

-$\mathbf{\sigma}_q$ is actually a scalar in the diagonal case

-Section 3.1: $\eta$ is random. Thus, the statement ``$\eta_j = O(1)$ does not make sense without some probabilistic condition.

-In Equation 9, you take absolute value of vector-valued quantities. This does not make sense. Also, since these are random variables, you need some probabilistic condition.

-The claim that as $\sigma_q$ shrinks, $g_s$ becomes small relative to $g_m$ is not precisely stated. Please state this precisely and explain clearly why it is true. Also, you don't address the gradient of the entropy term in your current discussion around Equation 9.

-Aligned gradients: please explain why ``it comes at the cost of increased $\ell_2$-norm of the full gradient.'' This is an important point that needs to be clearly justified.

-DP of Algorithm 1: Although the post-processing argument seems correct, a rigorous proof that Algorithm 1 is DP should be provided somewhere in the main body or appendix.

-Theorem 3.1: The first sentence is not clear/precise. Also, the claim before Theorem 3.1 that the variance is ``reduced'' is not quite established by Theorem 3.1 since the inequality in Equation 15 may not be strict. Either strengthen (15) to a strict inequality if possible, or weaken the claim before Theorem 3.1.

-Section 3.2 needs major improvements. The purpose of the section is unclear at the outset. Even the title of the section is not very informative. Clarify what the section aims to show or accomplish at the beginning of the section. Quantification of uncertainty (e.g. by DP confidence intervals) would be interesting, but it doesn't seem that you do that. Instead, you seem to aim to reduce the variance of your estimator via iterate averaging.

-Experiments: can you add plots showing error vs. epsilon for DPVI?

-experiments: In Figure 1a, where is the initial value of $\sigma_q$? I only see a plot comparing error in estimating mean vs. error in estimating std (which, by the way, is not necessarily the most natural or informative plot to include as your first plot).

-Section 4.3: logistic regression is usually thought of as a machine learning problem, not a VI problem. Please clarify why this is a VI problem. Is he ``Vanilla'' baseline just DP-SGD? Please clarify

---

> ### Author Response · Authors · 2023-06-09
> **Response to Reviewer E9ZR**
>
> **General improvements to the text**: We thank you for your careful reading of the paper. We have now improved the writing in the parts you pointed to in your review. Please let us know if something still remains unclear in the revised version.
>
> **“The problem of DP VI should be defined early in the introduction“ and “explanation/motivation/intuition of ELBO”**: We have now revised the Section 2.2. to more thoroughly discuss the background on ELBO. We hope that this also makes it more clear what the optimization target is in VI and hence in DPVI.
>
> **Additional references (gradient clipping with convex DP-SGD)**: We have added a sentence about these works in the related work.
>
> **Additional references (iterate averaging)**: Thank you for pointing us to these references on the use of iterate averaging in DP-SGD! We have now added the two first references to the Section 3.2, where we discuss the use of iterate averaging for our optimization task. Furthermore, we want to briefly summarize here the difference between these methods and our work. The iterate averaging in Bassily et al. 2014 seems to follow from the learning rate adaptation. Whereas in Bassily et al. 2019 and Lowy et al. 2021, the iterates obtained during the DP-SGD learning process are averaged in the end, which corresponds to the standard iterate averaging scheme. While both of these methods use iterate averaging to reduce DP induced noise, our method aims to actively search for the point of convergence in each parameter dimension, and starts the iterate averaging only after. The convergence is required for estimating the DP induced noise from the OU process we discussed earlier in Section 3.2.
>
> **"$\sigma_q$ is actually a scalar in the diagonal case"**: We believe that there has been a misunderstanding about what we mean with the diagonal covariance matrix. The diagonal covariance means that we estimate the posterior std. for each of the model parameters, and treat the parameters as independent. Thus the $\sigma_q$ is not a scalar in the diagonal case, but a vector of size of the model parameter’s dimension. The scalar $\sigma_q$ would correspond to the _spherical_ covariance matrix.
>
> **Section 3.1, Eq. 9**: Please see the response to all reviewers.
>
> **“You don't address the gradient of the entropy term in your current discussion around Equation 9”**: Our focus for the disparate scaling is in the data dependent parts of the gradient. Hence we have dropped the entropy term (which does not depend on the data) from gradient magnitude scaling. However, if you consider the entropy contribution for the gradients, a simple calculation shows that when using any of the transformation functions we consider, the entropy gradients are less than or equal to 1 in all dimensions. This means that, especially when we are far from the posterior mean, the mean gradients tend to be much larger compared to these, and thus the data dependent part of $g_s$ has a much more significant effect on the total gradient than the entropy. We have improved the discussion about this in the revised text.
>
> **“The claim that as $\sigma_q$ shrinks, $g_s$ becomes small relative to $g_m$ is not precisely stated”**: You are correct that the discussion about this needed some improvements. First of all, we have now added the probabilistic condition for the $\eta$ as outlined in our common response. Second, we have revised the text to make it more clear that we focus on the data dependent part of the $\mathbf{g}_s$, which does become small relative to $\mathbf{g}_m$ as the $\sigma_q$ shrinks as shown by Equation (12).
>
> **"Aligned gradients: please explain why ``it comes at the cost of increased $\ell_2$-norm of the full gradient.''**: This is an important point that needs to be clearly justified. This assertion refers to the preconditioning of gradients that we discussed just before introducing aligned gradients. That approach would simply scale the gradients $\mathbf{g}_s$ to match the magnitude of $\mathbf{g}_m$. As that would then increase the norm of $\mathbf{g}_s$ while keeping $\mathbf{g}_m$ fixed, the norm of the combined total gradient naturally increases. To avoid possible confusion on which approach this refers to, we have moved this discussion to the end of the paragraph “Preconditioning of Gradients”.
>
> **DP of Algorithm 1**: You are correct that the privacy guarantees for Algorithm 1 follow from the post-processing immunity. We have now added Corollary 3.1 in the revised text to make the claim more explicit. We also added the mathematical formulation of the post-processing immunity property of DP in Section 2.1.

---

> > ### Author Response · Authors · 2023-06-09
> > **Response to Reviewer E9ZR**
> >
> > **"Theorem 3.1: The first sentence is not clear/precise. Also, the claim before Theorem 3.1 that the variance is ``reduced'' is not quite established by Theorem 3.1 since the inequality in Equation 15 may not be strict."**: Thanks for pointing this out. We were able to make the calm of Theorem 3.1 stronger by assuming an upper bound for the derivative of transformation $T$. If we assume $T’(s) \leq 1$ for all s (which is the case for example with softplus transformation), we can establish a strict inequality for the variance comparison. Please also see the response to all reviewers.
> >
> > **"Section 3.2 needs major improvements"**: We believe there must have been a misunderstanding about the aims of this section. Our aim is to use the available parameter traces to estimate the DP induced noise for the final estimates. We model the parameter traces near convergence using the OU process, and use the iterate averaging to find the mean for this distribution. As the random-walk induced distribution is a Gaussian in this case, we can easily combine its variance with the variance of our VI posterior which is also a Gaussian. We have updated the Section in the revised version.
> >
> > **“Can you add plots showing error vs. epsilon for DPVI?”**: Figure 5 shows the effects of varying epsilon on the error in the variational parameters for all DPVI variants (vanilla and our proposed variants) for the logistic regression posterior on the Adult data set. Figure 3 also considers the effect of epsilon for the experiments with UK Biobank data.
> >
> > **"Figure 1a, where is the initial value of $\sigma_q$?**: This Figure shows the errors for mean and standard deviation parameters for multiple different initial values of $\sigma_q$. The point of the figure is to demonstrate that there is a trade-off between these accuracies: for some initial choices of $\sigma_q$ you obtain smaller errors for the mean parameters and for some you learn better the std parameters. Most importantly this Figure shows that the aligned method improves the trade-off compared to the other methods. The values used were $\sigma_q \in \{0.01, 0.032, 0.1, 0.316, 1\}$, and we have added these in the text.
> >
> > **"Please clarify why logistic regression is a VI problem. Is the ``Vanilla'' baseline just DP-SGD?"**: In our experiment we are interested in learning the posterior distribution of the weight parameter for logistic regression. Since there is no analytically tractable posterior for this model, we need to use approximate Bayesian inference methods such as VI for this problem. As in the preceding discussion and experiments, we use the term “vanilla” to refer to the original DPVI algorithm without any of our proposed gradient modifications.

---

### Review · Reviewer_bK6Z · 2023-06-01

**Summary Of Contributions:**

This paper studies the problem of learning a Gaussian variational posterior with differential privacy. They propose a version of DP SGD to learn the mean and variance of the Gaussian posterior which they call "aligned DPVI". Rather than use the Gaussian mechanism to add spherical noise to the gradient wrt both the mean and standard deviation, they notice that the gradient wrt the standard deviation can be written as a function of the gradient wrt the mean. They propose using the Gaussian mechanism to release the gradient wrt the mean, then post-process this into an estimate of the gradient wrt the standard deviation. They also compare to a version that scales the gradient wrt standard deviation so its scale matches the gradient wrt the mean, then adds Gaussian noise to both. They perform a series of experiments to show that these three algorithms behave as one would expect (aligned outperforms both other mechanisms).

**Audience:**

Yes

**Claims And Evidence:**

Yes

**Requested Changes:**

Discussion of Figure H.3 discussing under what circumstances the aligned algorithm can be worse than the vanilla algorithm.

**Strengths And Weaknesses:**

The paper is well-written and easy to follow. The experimental results are thorough, and the authors explore a variety of improvements over the vanilla algorithms (including iterate averaging and natural gradients). The paper offers an easy to implement improvement over the vanilla Gaussian mechanism for the problem of interest.

The key observation of this paper seems to be that that
	1. if you want to privately estimate 2 statistics, and one is a function of the other, then you shouldn’t spend privacy budget learning both, and
	2. for the case of variational inference with a Gaussian posterior, the derivative wrt the standard deviation is a function of the derivative wrt the mean.
The first fact is not very surprising. The proof of the second is short, but (given that the authors do not cite prior work for this result) does seem to be new to this work. The experimental results (excluding Figure H.3) behave as one would expect in that the “aligned” version of the algorithm uniformly outperforms the other methods in estimating both the mean and standard deviation.

I was surprised by the swapping of the order of the aligned and vanilla versions of natural grad in estimating the standard deviation is Figure H.3. The authors mention this is possibly due to hyper-parameter tuning? I would have liked to hear more about this since other than this graph, it seems like the aligned version is uniformly better than the vanilla making the decision of whether to use it in practice very simple.

Minor: On page 1 the authors refer to global sensitivity and local sensitivity as (sensitivity over all variables) and (sensitivity wrt a single variable). This is not the standard definition of these terms in differential privacy so the authors may want to consider alternate terminology.

Minor: Is the second last paragraph in the discussion referring to Figure H.3? If so, a pointer would be helpful.

---

> ### Author Response · Authors · 2023-06-09
> **Response to Reviewer bK6Z**
>
> **Discussion of Figure H.3**: Thanks for pointing this out, we agree that the presentation here could be improved. Figure H.3 shows indeed a curious swapping of order in error between natural gradients and aligned natural gradients for the standard deviation variational parameters. Note that this swap is not present in the location/mean variational parameters. This swap is not present in the left side of Figure H.4, which repeats the same experiment with the clipping threshold for both of these variants set to $0.02$: Here we see that the error in standard deviation parameters is the same for both, while the error in location parameters favors the aligned natural gradient variant, which is additionally much lower than in Fig H.3. These observations are in line with expectations - the poorer performance of natural gradients on location parameters is theorized in Section 3.1.1. In this experiment, aligned natural gradients can somewhat mitigate this effect but are still outperformed by plain aligned gradients. The discrepancy between the two figures is then likely to be due to a poor choice of C in Fig H.3 resulting in adding two much noise to converge within the given number of epochs. This highlights yet again the importance of choosing C appropriately. To clarify the presentation in Appendix H, we have swapped Fig H.3 and the left side of Fig H.4 and incorporated the points made above in the surrounding discussion in the revised version.
>
> **“Is the second last paragraph in the discussion referring to Figure H.3?”**: Yes it is. The wording we used originally might have been too negative towards the results of our hyperparameter tuning. As discussed above, we experimented with smaller clipping thresholds for the NG variants and obtained some improvements over the Figure H.3 where we used the same clipping threshold selection heuristic as for the aligned and vanilla approaches. Given the earlier discussion about Figure H.3, we have changed the word “without success” to “with limited success” in the revised version. For future work, it would be interesting to perform a more extensive hyperparameter tuning including for example the learning rate and number of iterations as the tunable parameters, to see if the NG methods make some non-trivial trade-offs in the learning that differ from the aligned and vanilla methods.
>
> **Alternate terminology for global and local sensitivity:** We are now using “total sensitivity” and “variable-specific sensitivity”.

---

### Author Response · Authors · 2023-06-09
**Response to all reviewers**

We thank the reviewers for their careful assessments of our work. We appreciate that reviewers have recognized that our experiments are thorough and demonstrate that the proposed solution is effective. We also appreciate the assessment that our paper is easy to read and that the message of our paper is clear. We will address individual comments in a dedicated response for each reviewer. We have also included a revised version of the paper with highlights on the major changes. Below, we will address comments made both by the reviewers E9ZR and n9qi.

**Probabilistic condition for the gradient magnitude discussion**: Reviewers E9ZR and n9qi noted that our gradient magnitude comparison in Section 3.1. was missing a probabilistic condition. We have now solved this in the revised version. Additionally we restated the claim to focus on a single dimension of the gradient vector at a time, making the use of absolute values justified.

**Making Theorem 3.1 more precise**: Reviewers E9ZR and n9qi pointed out that the initial conditions in Theorem 3.1 were a bit too vague. Reviewer E9ZR additionally suggested making the variance inequality strict. We have now revised the theorem accordingly. To summarize, we were able to make the variance inequality strict for gradients for which the $||g_s|| \neq 0$ by assuming $T’(s) \leq 1$ for the std. parameter transformation $T$. The case $||\mathbf{g}_s|| \neq 0$ would be almost surely satisfied, and we would have $||\mathbf{g}_s|| = 0$ if and only if the std. gradient would be zero for _all_ the individuals of the batch. Also the assumption $T’(s) \leq 1$ is satisfied by the softplus transformation which we used in all of our experiments.

---

### Decision · Action_Editors · 2023-08-21

**Recommendation:** Accept as is

**Comment:**

All the reviewers agree that the main concerns have been addressed, and the paper is technically solid. The paper is a great contribution to the differentially private variational inference problem. Thus I recommend acceptance.

**Audience:**

Yes

**Claims And Evidence:**

The claims are supported by good theoretical analysis and empirical results.